# Divergent regulation of basement membrane trafficking by human macrophages and cancer cells

Julian C. Bahr[1,2,5], Xiao-Yan Li[2,3,4,5], Tamar Y. Feinberg[2,3,4], Long Jiang[2,3,4] & Stephen J. Weiss ◎ [1,2,3,4] ✉

Macrophages and cancer cells populations are posited to navigate basement membrane barriers by either mobilizing proteolytic enzymes or deploying mechanical forces. Nevertheless, the relative roles, or identity, of the proteinase -dependent or -independent mechanisms used by macrophages versus cancer cells to transmigrate basement membrane barriers harboring physiologically-relevant covalent crosslinks remains ill-defined. Herein, both macrophages and cancer cells are shown to mobilize membrane-anchored matrix metalloproteinases to proteolytically remodel native basement membranes isolated from murine tissues while infiltrating the underlying interstitial matrix ex vivo. In the absence of proteolytic activity, however, only macrophages deploy actomyosin-generated forces to transmigrate basement membrane pores, thereby providing the cells with proteinase-independent access to the interstitial matrix while simultaneously exerting global effects on the macrophage transcriptome. By contrast, cancer cell invasive activity is reliant on metalloproteinase activity and neither mechanical force nor changes in nuclear rigidity rescue basement membrane transmigration. These studies identify membrane-anchored matrix metalloproteinases as key proteolytic effectors of basement membrane remodeling by macrophages and cancer cells while also defining the divergent invasive strategies used by normal and neoplastic cells to traverse native tissue barriers.

Macrophages as well as cancer cells can individually or cooperatively infiltrate and remodel the extracellular matrix (ECM) of native tissues[1–6]. In (patho)physiologic states, both macrophages and carcinoma cells confront at least one of two distinct ECM barriers, i.e., the basement membrane or the interstitial matrix[1–7]. As a specialized form of ECM, the basement membrane subtends all epithelial and endothelial cell layers[4,6,8]. Despite ranging in thickness from only 50–400 nm, basement membranes are mechanically rigid barriers in almost all tissues, largely owing to a covalently cross-linked network of tightly intertwined type IV collagen fibers that non-covalently associate

with a laminin meshwork as well as a complex mix of 30 or more other matrix components[8,9]. In turn, the underlying interstitial tissues are dominated by an interwoven composite of type I/III collagen, elastin, glycoproteins, proteoglycans, and glycosaminoglycans[5,7].

Upon interacting with the basement membrane–interstitial matrix continuum in vivo, current evidence suggests normal as well as cancer cells irreversibly or reversibly remodel ECM interfaces in order to drive tissue-invasive activity[4,6,10,11]. Irreversible changes in ECM structure are most frequently linked to proteolytic remodeling, but recent studies suggest that invading cells can also apply

[1]Cancer Biology Graduate Program, University of Michigan, Ann Arbor, MI 48109, USA. [2]Life Sciences Institute, University of Michigan, Ann Arbor, MI 48109, USA. [3]Division of Genetic Medicine, University of Michigan, Ann Arbor, MI 48109, USA. [4]Department of Internal Medicine, University of Michigan, Ann Arbor, MI 48109, USA. [5]These authors contributed equally: Julian C. Bahr, Xiao-Yan Li. ✉e-mail: sjweiss@umich.edu

mechanical forces that can precipitate the physical rupturing of basement membrane architecture[6,12,13]. By contrast, fully reversible changes in matrix architecture that support invasion programs have also been linked to changes in the mechanical properties of the motile cell's nucleus, its most rigid intracellular organelle[6,14,15]. To date, however, efforts to characterize macrophage– or cancer cell–ECM interactions, and the relative roles of proteinase- dependent versus independent processes, have largely been confined to the use of artificial matrix constructs that lack the critical structural organization and mechanical properties that characterize native ECM structures assembled in vivo[4,8,9,16–23]. Hence, despite the fact that both macrophages and cancer cells can mobilize a complex repertoire of proteolytic enzymes while exerting mechanical forces in and outside the cell, which, if any, of these proteolytic or non-proteolytic systems participate in the remodeling and transmigration of native basement membranes remains the subject of debate[5,6,10,24–28].

To define the molecular mechanisms that underlie macrophage-dependent versus cancer cell-mediated ECM remodeling, we have examined interactions between human macrophages or cancer cells as they confront native basement membrane-interstitial matrix explants. Using live cell confocal imaging, we find that macrophages and cancer cells are both able to penetrate native basement membrane barriers and infiltrate the underlying interstitium by mobilizing membrane-anchored matrix metalloproteinases. However, only macrophages are able to alternatively use actomyosin-dependent mechanical forces to drive an invasion program that operates independently of matrix-degradative activity by deforming cell shape to penetrate pre-existing basement membrane pores that prove inaccessible to cancer cell trafficking. Further, we find that the selection of proteinase–dependent versus –independent invasion programs exerts major effects on macrophage transcriptional programs, thereby providing insights into the divergent mechanisms used by macrophages and cancer cells to negotiate basement membrane-interstitial matrix barriers.

## Results

### Primary human macrophages remodel native basement membrane

Using decellularized mesenteric sheets[29], three-dimensional (3D) reconstructions of immunofluorescent and second harmonic generation images allow visualization of a reflected basement membrane bilayer that unsheathes an intervening interstitial matrix (Fig. 1a, b and Supplementary Movies 1 and 2). En face confocal images of laminin- or type IV collagen-stained tissues highlight the sheet-like architecture of the basement membrane, while orthogonal *xz* and *yz* reconstructions permit visualization of the apical and basal basement membrane layers that are separated by the ~50 μm-thick (unstained) interstitial matrix (Fig. 1c).

As mechanical integrity of the basement membrane is largely defined by a variable number of intermolecular sulfilimine bonds formed between the C-terminal domains of opposing type IV collagen trimers (Fig. 1d)[7], we assessed the relative frequency of these covalent cross-links in isolated basement membranes. Following digestion with bacterial collagenase, the triple-helical, C-terminal domains of type IV collagen molecules (termed NC1 domains) remain associated as either non-covalently or covalently associated dimers (i.e., the NC1 dimers contain a total of six type IV collagen chains)[30,31]. Using this approach, the type IV collagen network in basement membrane explants is shown to be dominated by covalent cross-links (Fig. 1e) at levels similar to those found in other highly cross-linked basement membranes (78% dimer/22% monomer)[32], thereby confirming that tissue-invasive cells confront a physiologically-relevant barrier.

Carboxyfluorescein diacetate succinimidyl ester (CFSE)-labeled human monocyte-derived macrophages were next cultured atop basement membranes pre-labeled with fluorescently tagged anti-laminin antibodies in the presence of $F_c$ receptor blocking reagents to prevent direct interactions between the macrophages and the antibody-coated surface[33,34]. After 48 h, the macrophage-tissue constructs were imaged for 160 min using real-time spinning disc confocal microscopy. As shown, macrophages (green) are found adherent to the basement membrane (red) in association with the appearance of distinct 5–10 μm diameter perforations in the labeled matrix (Fig. 1f, arrows). While real-time imaging of macrophage surface contours over this timespan detects only small changes in lateral spreading (Fig. 1g), the basement membrane surface is actively penetrated by cellular protrusions (Fig. 1h and Supplementary Movie 3). Under higher magnification, real-time imaging of a single basement membrane pore in association with an overlying macrophage reveals an increase in perforation size from ~11 to ~17 μm$^2$ (Fig. 1i), a finding consistent with the active remodeling of the cell–matrix interface. Hence, within 48 h of culture, human macrophages remodel native basement membrane barriers while breaching the surface with invasive membrane protrusions.

### Immune-polarizing stimuli alter the basement membrane remodeling potential of human macrophages

Given that macrophages serve discrete functions during the initiation and resolution of inflammatory responses[3], we sought to characterize the effect of immune polarizing stimuli on basement membrane remodeling. As expected, macrophages stimulated with *E. coli* lipo-polysaccharide (LPS) upregulate *TNFα* and downregulate *MRC1* transcript levels[35,36] (Fig. 2a). Conversely, polarizing macrophages with the cytokine, IL-4, downregulates *TNFα* and upregulates *MRC1* transcript levels (Fig. 2a)[35,36]. As such, unstimulated or variably polarized macrophages were cultured atop the tissue constructs for a 6-day culture period. Under basal conditions, macrophages generate large numbers of ~10 μm diameter perforations with basement membrane 'holes' constituting approximately 15% of the total surface area (Fig. 2b, c). LPS-polarized cells likewise remodel the basement membrane, but the percent surface area perforated increases ~2-fold as does the average size of the perforations (Fig. 2b, c). While IL-4-dependent polarization can be linked to a tissue-remodeling phenotype under select conditions[37,38], these macrophages remodel basement membranes to a degree similar to that observed with unstimulated macrophages (Fig. 2b, c).

Given that LPS-stimulated macrophages mount the most robust remodeling program, we used these cells to characterize the nature of the basement membrane perforations. As basement membrane remodeling has been proposed to be generated as a function of either reversible mechanical distortions or frank proteolysis[6,10], we assessed basement membrane structure by SEM following the 6-day culture period. As shown, clearly demarcated ~10 μm diameter perforations are found in macrophage-exposed, but not control, constructs (Fig. 2d). In tandem with the imaging of basement membrane denudation, type IV collagen fragments are solubilized from the matrix surface (Fig. 2e), thereby supporting the activation of a proteolytic program. Consistent with these findings, normalized fluorescence intensity profiles of macrophage-mediated perforations indicate that basement membrane proteins are removed completely from the cleared zones rather than mechanically displaced to the sides of the defects (Fig. 2f).

While recent studies have highlighted the ability of human and mouse macrophages to respond to specific inflammatory stimuli in transcriptionally and phenotypically distinct fashion[35,39], we find that mouse bone marrow-derived macrophages also perforate the basement membrane in response to LPS or IL-4 polarization (Fig. 3a). Under these conditions, LPS-

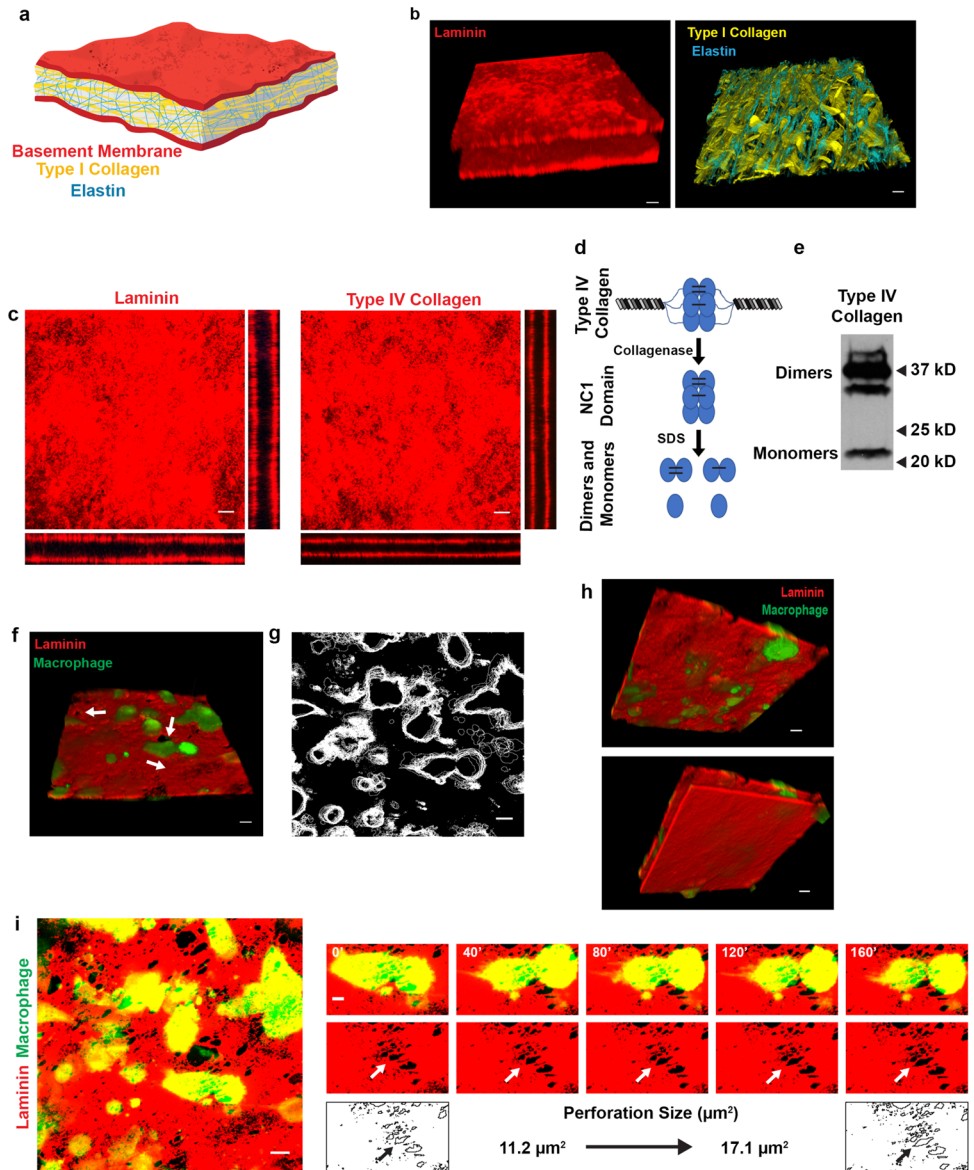

**Fig. 1 | Human macrophage interactions with native basement membrane explants. a** Schematic illustration of the mesentery extracellular matrix. **b** 3D confocal reconstructions of laminin (red; left panel) and elastin (blue; right panel), with second harmonic generation of type I collagen (yellow; right panel) in mesentery explants. **c** En face and orthogonal immunofluorescence of laminin (red) and type IV collagen (red). Results representative of 5 independent experiments. **d** Schematic of type IV collagen dimer–monomer content analysis. After collagenase digestion of type IV collagen, the hexameric NC1 domain remains intact. The hexamer can be dissociated via non-reducing SDS-PAGE into sulfilimine-cross-linked dimers and non-cross-linked monomers. **e** Type IV collagen dimer–monomer content analysis as determined by western blotting. Image is representative of two experiments performed. **f** 3D confocal reconstruction of human macrophages (green) atop the apical face of a basement membrane (red) with adjacent perforations (arrow) after 48 h. Results representative of five independent experiments performed. **g** Overlay of macrophage outlines captured every 10 min for 160 min. Results representative of 3 independent experiments performed. **h** 3D reconstruction from **f** rotated 180° showing the bottom face of the apical basement membrane (top panel with laminin colored red) with penetrating macrophage protrusions (green) highlighted as well as the bottom face of the basal basement membrane (bottom panel). Results representative of two independent experiments performed. **i** Immunofluorescence of the apical basement membrane layer (red) and macrophages (green; left panel) with a macrophage actively expanding a perforation in the basement membrane (small panels, arrows) over a 160 min time period. Bars: 20 μm (**b**, **f**, **h**); 10 μm (**c**, **g**, **i** left panel); 5 μm (**i** right panel). Results representative of seven experiments performed for **i**. Source data are provided as a Source data file. All figure panels containing red/green combined images have been separated and can be found in Supplementary Data 2.

stimulated mouse macrophages remodel an area approximately three times larger than control or IL-4 stimulated cells without a significant change in perforation size (Fig. 3c). Interestingly, mouse multinucleated giant cells are occasionally formed in response to IL-4[40], but they express only minimal basement membrane remodeling activity (Fig. 3b). Taken together, these data demonstrate that human as well as mouse macrophages can proteolytically remodel native tissue barriers via processes responsive to immune polarization.

## Macrophage polarization and basement membrane remodeling correlate with proteinase expression

As both mouse and human macrophages display similar matrix-remodeling phenotypes, we first used mouse macrophages as a genetically modifiable system to identify the underlying mechanisms responsible for basement membrane remodeling. To this end, we transcriptionally profiled mouse macrophages after a 24 h culture period under either basal, LPS-stimulated or IL-4-stimulated conditions. As expected, the upregulation of mouse-specific polarization

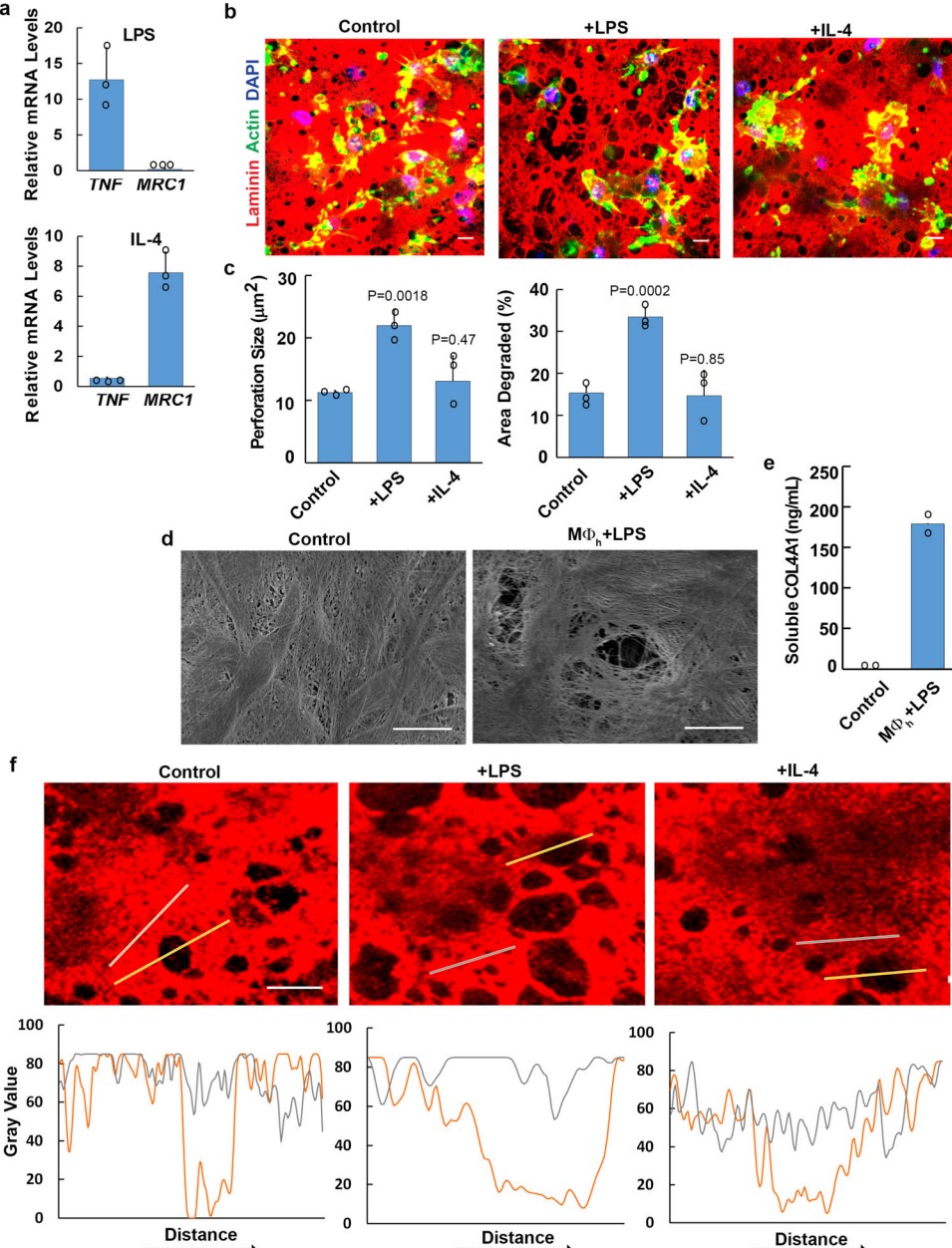

**Fig. 2 | Polarized human macrophage-dependent remodeling of basement membrane explants. a** Transcript expression of immune response genes as analyzed by qPCR in human macrophages polarized with LPS (1 μg/mL) or recombinant human IL-4 (20 ng/mL). Results are expressed as mean fold-change relative to control ± SEM (*n* = 3 independent exps). **b** Basement membrane laminin immunofluorescence following a 6 d culture with control, LPS- (1 μg/mL) or recombinant IL-4- (20 ng/mL) treated macrophages. Images shown are representative of three independent experiments. **c** Quantification of basement membrane perforation size or area degraded as analyzed by ImageJ pixel analysis of each condition in **b**. Results are expressed as mean ± SEM (*n* = 3 independent exps) with significance determined by two-tailed *t* test. **d** Scanning electron micrograph of basement membrane stripped of cells either after culture with medium alone or with LPS-polarized human macrophages (MØh) for 6 days. Results representative of two independent experiments performed. **e** Quantification of soluble type IV collagen detected in cell-free media on day 3. Results are expressed as mean of two independent experiments performed. **f** Normalized fluorescence intensity profiles of laminin (red) across non-degraded (gray lines) versus degraded basement membrane perforations (orange lines) in explants cultured with control, LPS-, or IL-4-treated macrophages for 6 days. Results representative of four independent experiments. Bars: (**b–d, f**) 10 μm. Source data are provided as a Source data file.

markers, *Nos2* and *Arg1*[36,41], correlated with LPS and IL-4 stimulation, respectively (Fig. 3d). In addition, a large number of proteases belonging to the metalloproteinase, cysteine, aspartic, and serine proteinase/receptor families previously associated with ECM remodeling are expressed under these conditions (Fig. 3e)[17,37,42,43]. Of note, however, only a small number of these proteases are differentially expressed in response to LPS or IL-4, with a distinct subset of these enzymes altering their transcript levels in a pattern that correlated with the matrix-remodeling phenotype, including the metalloproteases,

*Mt1-mmp/Mmp14* and *Adamts4*, the serine proteases, *Htra4* and *Ctrl*; and the serine protease receptor, *Pluar* (Fig. 3f).

## Matrix metalloproteinases are required for basement membrane remodeling
Cognizant of the fact that correlative changes in transcript levels do not necessarily correlate with matrix degradation activity, we next sought to identify effector proteases responsible for matrix remodeling by culturing mouse macrophages atop tissue explants in the

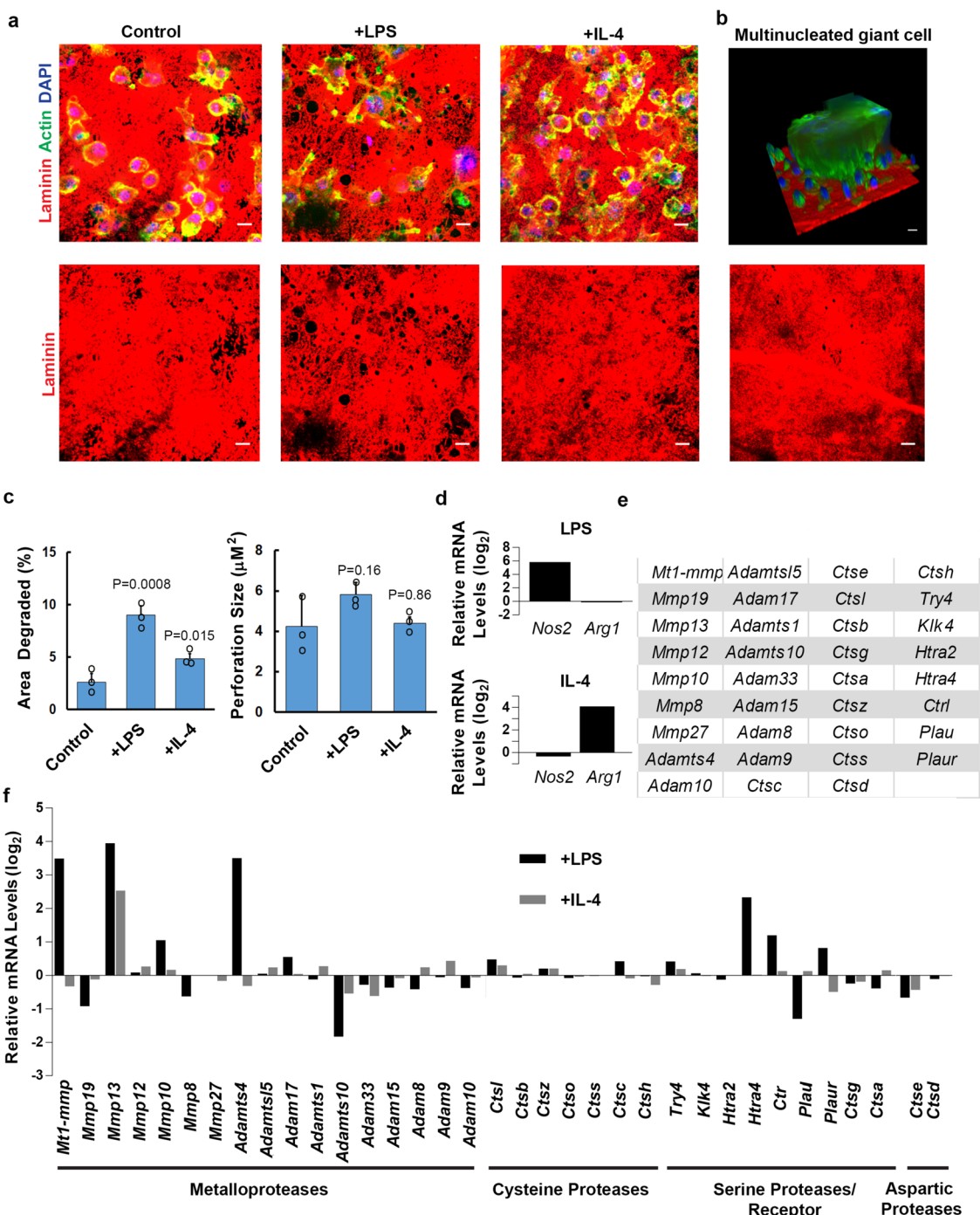

**Fig. 3 | Polarized mouse macrophages express a suite of proteases. a** Basement membrane laminin immunofluorescence following culture with mouse macrophages for 6 days in the presence of media alone, LPS (1 μg/mL), or recombinant mouse IL-4 (20 ng/mL). Images shown are representative of three independent experiments. **b** 3D, en face and orthogonal images of a multinucleated giant cell formed atop basement membrane explants in response to IL-4. Results representative of two independent experiments performed. **c** Quantification of basement membrane area degraded and basement membrane perforation size as analyzed by ImageJ pixel analysis of each condition from **a**. Results are expressed as mean ± SEM ($n$ = 3 independent experiments) with significance determined by two-tailed $t$ test. **d** Transcript expression for two biological replicates of mouse macrophages left unstimulated, polarized with LPS (1 μg/mL), or polarized with recombinant mouse IL-4 (20 ng/mL) for 24 h. Relative mRNA expression levels of mouse-specific immune response genes (**d**), proteases or protease receptor with an absolute gene expression value of at least $2^4$ (**e**), and the relative expression of those proteases/receptor in response to LPS or IL-4 (**f**) are presented. **d, f** are on a log₂ scale. Results are presented as the mean of two independent experiments. Source data are provided as a Source data file.

presence of broad-spectrum inhibitors directed against cysteine, serine, or metalloprotease family members[20,29,42,44,45]. Despite the expression of multiple proteases by LPS-stimulated mouse macrophages, the addition of high concentrations of validated cysteine or serine protease inhibitors[46,47] fail to inhibit basement membrane remodeling to a significant degree (Supplementary Fig. 1). In

contrast, the pan-specific matrix metalloprotease (MMP) inhibitor, BB-94[48,49], significantly blocks basement membrane degradation (Supplementary Fig. 1). To further narrow the number of candidate proteases, we took advantage of the fact that endogenous protease inhibitors, known as tissue inhibitor of metalloproteinases (TIMPs), can be used to preferentially block the proteolytic activity of

secreted versus membrane-anchored MMPs[29,50,51]. In the presence of TIMP-1, a more specific inhibitor of secreted MMPs[11,51], the remodeling program is unaffected (Supplementary Fig. 1). By contrast, TIMP-2, an endogenous inhibitor of both secreted and type I membrane-anchored MMPs[11,29,51], abrogates basement membrane degradation (Supplementary Fig. 1). As BB-94 and TIMP-2 are the only inhibitors that effectively block basement membrane degradation, these results indicate that a membrane-type MMP is likely the sole protease required for basement membrane remodeling.

## MT1-MMP is the dominant effector responsible for macrophage-mediated remodeling of the basement membrane

While at least four members of the MT-MMP family are sensitive to TIMP-2 (i.e., MT1-MMP, MT2-MMP, MT3-MMP and MT5-MMP)[5,51], transcriptional profiling of LPS-stimulated mouse macrophages identified *Mt1-mmp* as the sole membrane-anchored MMP expressed under these conditions (Fig. 3e, f). Given that the increase in *Mt1-mmp* transcript levels most closely correlated with the basement membrane remodeling phenotype, we confirmed by immunostaining permeabilized macrophages cultured atop the basement membrane that the proteinase is upregulated following polarization with LPS and, to a lesser degree, IL-4 (Fig. 4a). As such, to define the impact of Mt1-mmp activity on the matrix-remodeling program, mouse macrophages were prepared from *Mt1-mmp*$^{-/-}$ mice and cultured atop the tissue explants. Underlining an essential requirement for Mt1-mmp in basement membrane remodeling, *Mt1-mmp*$^{-/-}$ macrophages fail to display matrix-degradative activity under basal, LPS-, or IL-4- stimulated conditions (Fig. 4b, c) despite maintaining identical expression of more than 180 non-targeted cysteine, serine, aspartyl and metallo- proteases (Supplementary Data 1). Importantly, following transduction of *Mt1-mmp*$^{-/-}$ macrophages with an MT1-MMP/mCherry-tagged construct[52], basement membrane perforations materialize coincident with macrophages extending MT1-MMP/mCherry-positive protrusions into the underlying interstitial stroma (Fig. 4c, d).

To determine whether the mouse Mt1-mmp-dependent regulation of basement membrane remodeling can be extended to human cells, we assessed MT1-MMP mRNA, protein and surface expression in human macrophages. However, while LPS or IL-4 increased *MT1-MMP* transcript levels, protein expression remains largely unchanged from control cells (Fig. 5a). Nevertheless, as assessed by confocal imaging, though MT1-MMP was found to localize in permeabilized cells to the peri-nuclear ER/Golgi region as well as trafficking vesicles throughout the cell (Fig. 5b upper panels), surface-associated MT1-MMP levels increase in response to LPS polarization alone (Fig. 5b, lower panels and Fig. 5c). Consistent with these findings, when human macrophages are cultured atop the basement membrane in the presence of BB-94, or a monoclonal antibody directed against the catalytic domain of MT1-MMP[53–55], matrix degradation is almost completely ablated (Fig. 5d–f). Hence, Mt1-mmp/MT1-MMP is required for both mouse and human macrophage-mediated basement membrane degradation, respectively.

## Macrophages actively transmigrate the basement membrane–interstitial matrix interface independently of proteolysis

Coincident with basement membrane degradation, both orthogonal reconstructions of macrophage-explant cultures (Fig. 6a) as well as en face views of the underlying interstitial space (Fig. 6b) reveal that approximately 60% of the cells actively infiltrate the explant surface wherein entire cell bodies and nuclei are found below the apical basement membrane face (Fig. 6a–c). Of note, more than 30% of the tissue-invasive macrophages continue to invade after they perforate the apical face of the explant and traverse the opposing reflected basement membrane, demonstrating that the remodeling program occurs regardless of the symmetry of basement membrane proteins[56]

(Supplementary Fig. 2). Unexpectedly, however, when basement membrane proteolytic remodeling by human or mouse macrophages is blocked by targeting MT1-MMP/Mt1-mmp or inhibiting MMPs with BB-94, the cells continue to penetrate the basement membrane surface (Fig. 6d) where invasive structures are clearly seen on the reflected face of the basement membrane in the absence of widespread type IV collagen or laminin degradation (Fig. 6e). Further, when either MT1-MMP activity alone is blocked or macrophages cultured in the presence of a pan-specific proteinase inhibitor cocktail directed against MMPs as well as serine, cysteine and aspartyl proteases[45], cells continue to access the interstitial space (Fig. 6f) such that the number of basement membrane- transmigrating cells is similar to that observed under control conditions at the 6 d time point (Fig. 6c). Interestingly, however, at earlier times in the culture period (i.e., days 2, 3 and 4), control macrophages transmigrate more rapidly than MMP-inhibited cells, suggesting that proteinase-competent macrophages maintain a heightened invasive activity as a function of their ability to actively degrade the explant matrix (Supplementary Fig. 4).

## Basement membrane pores provide macrophages access to the stromal compartment

While macrophages are able to traverse tissue explants independently of detectable proteolytic remodeling, basement membrane "pores" have been identified in virtually all tissues that accommodate mesodermal–stromal contact–and possibly, myeloid cell trafficking[44,57–64]. As such, we considered the possibility that macrophages gain access to the interstitium through similar structures, thereby bypassing a proteolytic requirement[44,59–64]. Indeed, following imaging, the peritoneal basement membrane harbors a series of ~1 μm diameter pores before decellularization (Fig. 7a). Following decellularization, normalized fluorescence intensity profiles of laminin/type IV collagen likewise allow for the identification of potential passageways (Supplementary Fig. 3). To monitor the potential access of macrophages to these pore-like structures, tissue explants and human macrophages were fluorescently pre-labeled and transmigration captured by live imaging in the presence of the MT1-MMP-blocking antibody. Over a 7-hour time-course, orthogonal reconstructions illustrate vertical movement through the pores (Fig. 7b). Consistent with the non-proteolytic nature of the transmigration program, basement membrane pores widen as macrophages gain access to the underlying stromal compartment when viewed in cross section or en face (Fig. 7c, d) while fluorescence line scans of the generated pores highlight the buildup of excess laminin at the pore edges (Fig. 7e). In turn, following transmigration, the widened basement membrane pores undergo an elastic recoil over both a 4- and 24-h period (Fig. 7c, f). Changes in pore size are not observed in explants cultured without macrophages or in pores located at sites distant from adherent macrophages (Supplementary Fig. 3). Finally, when scanning over the entire surface of macrophage-explant co-cultures preformed in the presence of the proteinase inhibitor cocktail, quantitative analysis of basement membrane pore size over the 6-day culture period demonstrates a steady increase over the first 4 days of culture that then recedes by day 6 when the bulk of the transmigration process is near complete (Fig. 7g).

In other cell systems, non-proteolytic mechanisms of invasion have been linked to the transfer of mechanical forces from the cell body to either the surrounding matrix or the perinuclear compartment as a means to shape the rigid nucleus to a degree that allows small ECM pores to be negotiated[6,22,65–67]. In an effort to define the contribution of actomyosin-dependent contractility to invasion, macrophage transmigration into the interstitial matrix was assessed in the presence of the Rho kinase inhibitor, Y27632, or the myosin II inhibitor, blebbistatin[65–67]. In contrast to the ROCK-independent invasion used by *C. elegans* anchor cells to cross basement membranes in situ[13], the addition of either Y27632 or blebbistatin to either control or BB-94-supplemented cultures significantly blocks macrophage-basement

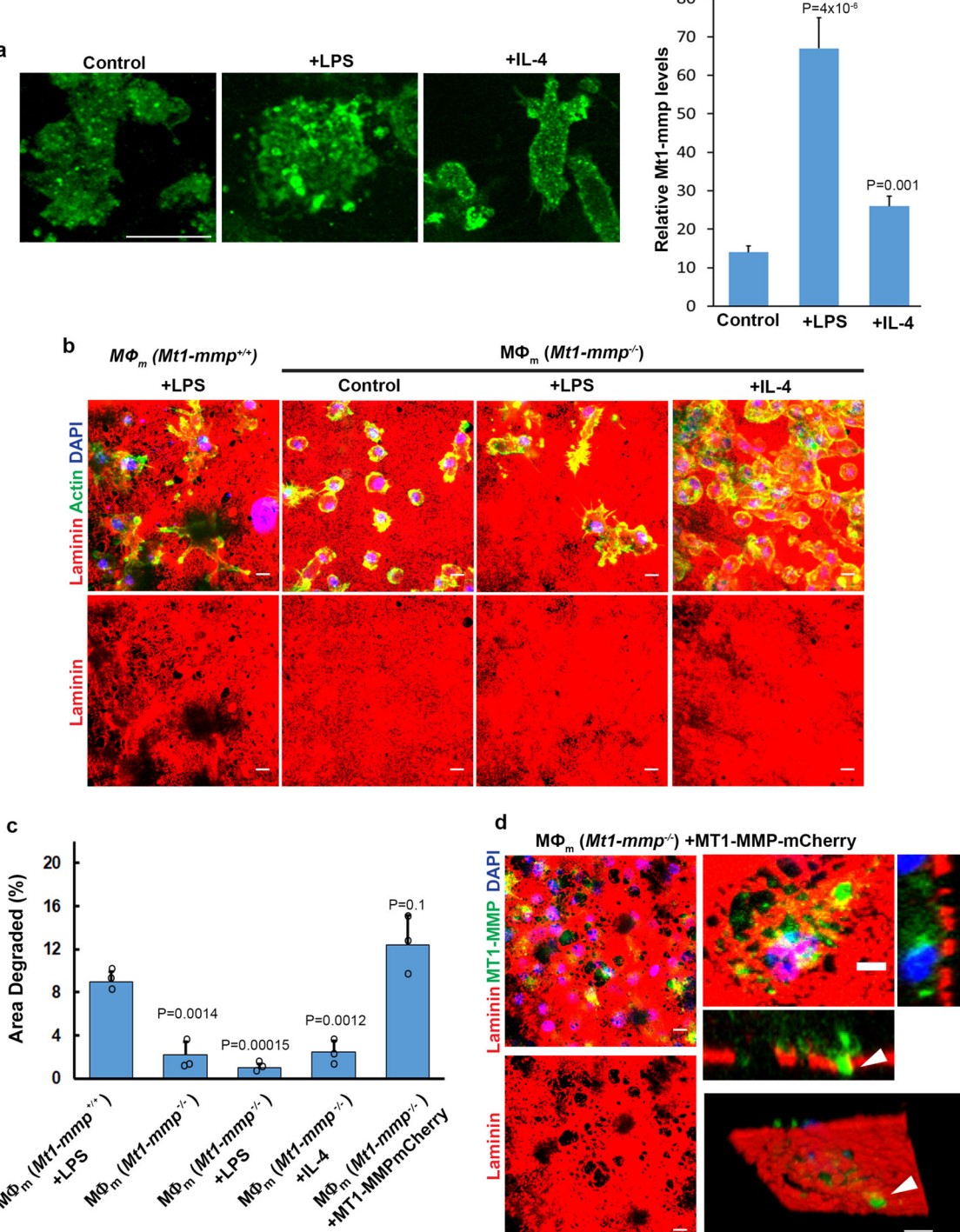

**Fig. 4 | Mt1-mmp-dependent mouse macrophage-mediated basement membrane remodeling. a** Mt1-mmp immunostaining (green) of mouse macrophages cultured on basement membrane explants (unstained) in control media, polarized with LPS (1 μg/mL) or recombinant mouse IL-4 (20 ng/mL) with relative immunofluorescence quantified. Results are representative of 3 experiments performed with results from a single experiment with Mt1-mmp quantified in 10 randomly selected cells as mean ± SEM with significance determined by two-tailed *t* test. **b** Laminin immunofluorescence of basement membranes cultured with LPS-polarized *Mt1-mmp*⁺/⁺ mouse macrophages (MØₘ) or unstimulated, LPS-, and IL-4-polarized *Mt1-mmp*⁻/⁻ mouse cells for 6 days. Results representative of three independent experiments performed.

**c** Quantification of the area of basement membrane degraded as analyzed by ImageJ pixel analysis under each set of conditions from (A). Results are expressed as mean ± SEM (*n* = 3 independent exps) with significance determined by two-tailed *t* test. Bars: 10 μm. **d** Laminin immunofluorescence of *Mt1-mmp*⁻/⁻ mouse macrophages transduced with a lentiviral MT1-MMP-mCherry vector (pseudo-colored green) for 48 h before culture on a basement membrane explant (pseudo-colored red) for 6 days. MT1-MMP-mCherry-positive protrusions are localized to basement membrane perforations (arrowheads) in orthogonal cross-sections or viewed en face. Images shown are representative of three independent experiments. Bars: left panels, 10 μm; right panels; 5 μm. Source data are provided as a Source data file.

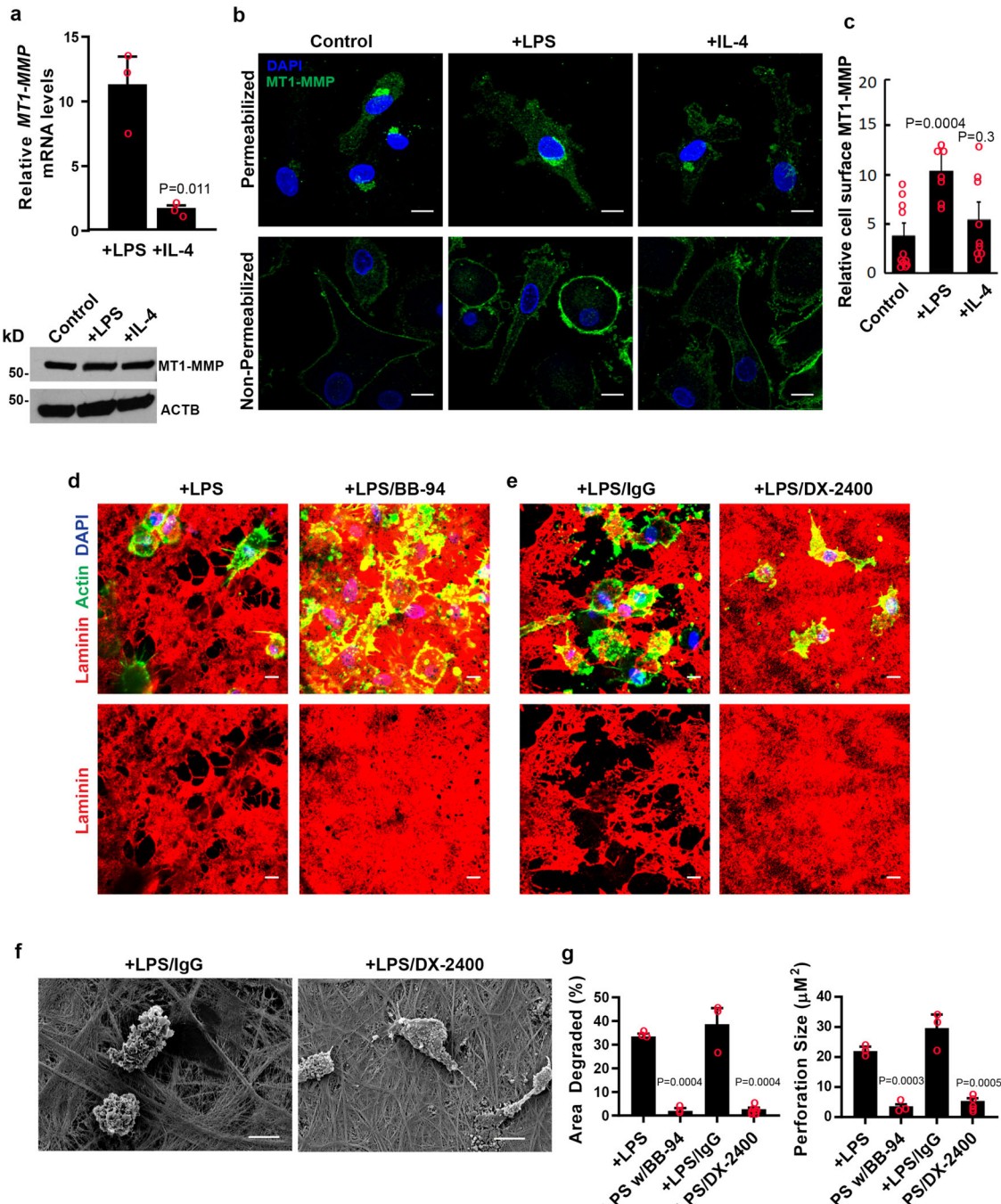

**Fig. 5 | Human macrophages mobilize MT1-MMP to degrade basement membranes. a** Relative *MT1-MMP*/MT1-MMP expression in human macrophages left unstimulated, polarized with LPS (1 μg/mL) or recombinant human IL-4 (20 ng/mL) as determined by qPCR (top panel) or western blot (bottom panel). Results expressed as mean ± SEM (*n* = 3 independent exps) with significance determined by two-tailed *t* test. **b**, **c** Confocal images of endogenous MT1-MMP immuno-fluorescence (green) in permeabilized (top three panels) or non-permeabilized (bottom 3 panels) human macrophages counterstained with DAPI (blue). In **c**, cell surface MT1-MMP immunofluorescence is shown from a single experiment of 3 performed where staining intensity in control (*n* = 11), LPS-stimulated (*n* = 9) and IL-4-treated (*n* = 9) cells and quantified as mean ± SEM with significance determined by two-tailed *t* test. **d**, **e** Basement membrane laminin immunofluorescence following culture with macrophages in the presence of LPS (1 μg/mL) without or with 5 μM BB-94, or 75 μg/mL IgG control antibody or 75 μg/mL of MT1-MMP blocking antibody, DX-2400, for 6 days (**e**). Results representative of three independent experiments performed. **f** Scanning electron micrograph of mesentery basement membrane after culture with macrophages in the presence of LPS (1 μg/mL) and either 75 μg/mL IgG or 75 μg/mL DX-2400 for 6 days. Images shown in **b**, **d**, **f** are representative of three replicates. Bars: **b**–**e** 10 μm. **g** Quantification of the area of basement membrane degraded and perforation size as analyzed by ImageJ pixel analysis of each condition from **d**, **e**. Results are expressed as mean ± SEM (*n* = 3 and *n* = 5, respectively, independent exps) with significance determined by two-tailed *t* test. Source data are provided as a Source data file.

membrane transmigration without affecting matrix degradation (Fig. 8a, b and Supplementary Fig 4), highlighting the importance of actomyosin-dependent forces in supporting proteinase-dependent or proteinase-independent invasion. Interestingly, in the presence of either inhibitor, macrophages retain the ability to insert cell protru-sions through the basement membrane pores while their nuclei remain confined to the upper surface of the basement membrane (Fig. 8c–e). Nevertheless, despite blocking macrophage transmigration, basement

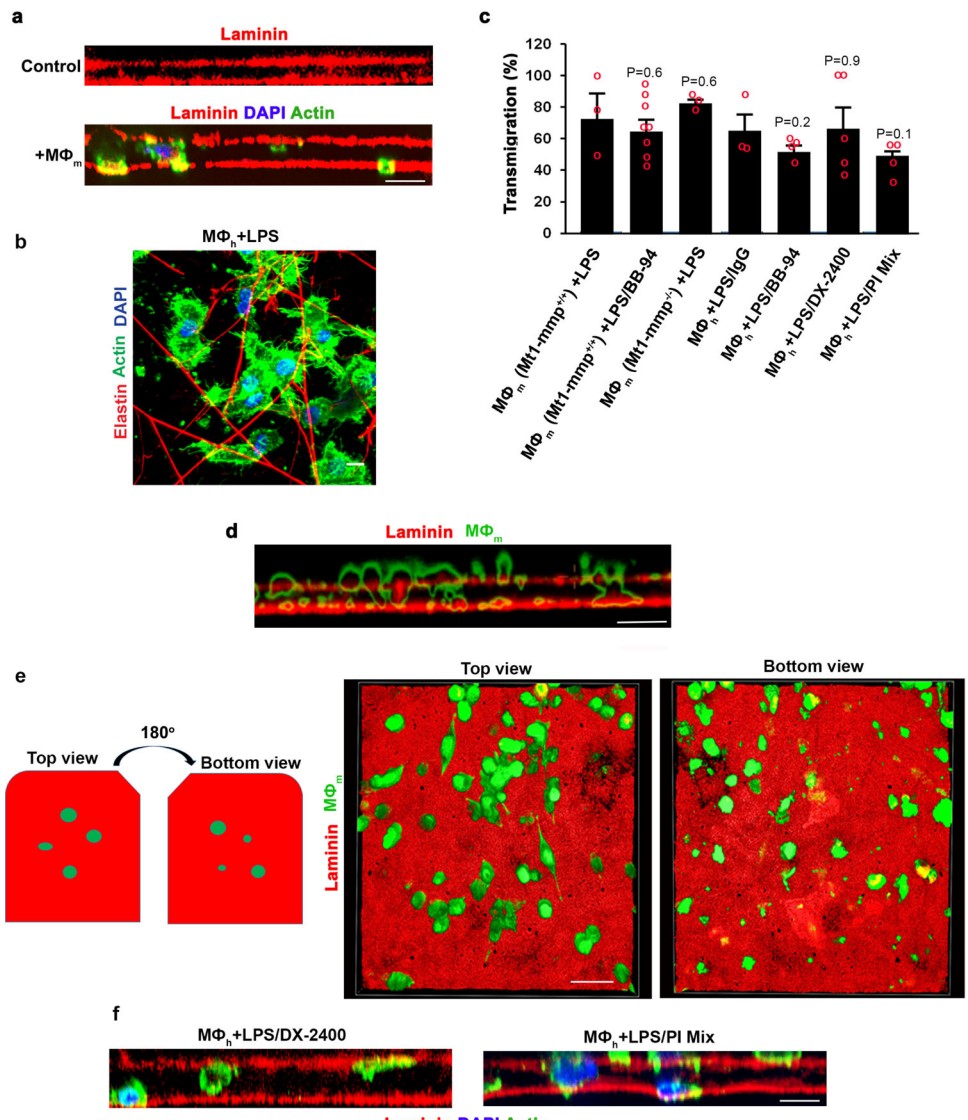

**Fig. 6 | Characterization of proteinase-dependent and proteinase-independent macrophage invasion programs. a** Orthogonal view reconstructions of laminin-stained basement membranes (red) following culture with LPS (1 μg/mL)-polarized *Mt1-mmp*⁺/⁺ mouse macrophages (nuclei and F-actin stained blue and green, respectively) for 6 days. Bar: 10 μm. Results representative of three independent experiments performed. **b** En face view of LPS-polarized *Mt1-mmp*⁺/⁺ mouse macrophages that traversed the apical face of the basement membrane and accumulated in the elastin-rich interstitium after a 6 d culture period. Bar: 10 μm. Results representative of three independent experiments performed. **c** *Mt1-mmp*⁺/⁺ or knockout mouse macrophages as well as human macrophages were cultured with LPS (1 μg/mL) in the absence or presence of 5 μM BB-94, the presence or absence of 75 μg/mL IgG, 75 μg/mL DX-2400, or a protease inhibitor mix (100 μM E-64d, 100 μg/mL aprotinin, 10 μM pepstatin A, 100 μg/mL SBTI, 5 μM BB-94, 2 μM leupeptin) for 6 days and transmigrated mouse (MØ$_m$) or human macrophages (MØ$_h$) located between the two basement membranes quantified as the percentage of the total number of cells. Results are presented as the mean ± SEM ($n$ = 3, 8, 3, 3, 4, 5, and 4 independent exps of each variable, respectively) with significance determined by two-tailed $t$ test. **d, e** Orthogonal (**d**) and 3D en face (**e**) reconstructions of laminin-stained basement membrane explants (red) following culture with LPS (1 μg/mL) -polarized *Mt1-mmp*⁺/⁺ mouse macrophages (green-stained with CFSE) in the presence of 5 μM BB-94 for 6 days. En face reconstructions show the upper and lower surfaces, respectively, of the apical basement membrane. Results representative of three independent experiments performed. Bars: 10 μm. **f** Orthogonal view reconstructions of laminin-stained basement membranes (red) following culture with LPS (1 μg/mL)-polarized human macrophages (nuclei and F-actin stained blue and green, respectively) cultured in the presence of 75 μg/mL DX-2400, or a protease inhibitor mix (described in **c**) for 6 days. Results representative of three independent experiments performed. Bar: 10 μm. Source data are provided as a Source data file.

membrane pores are enlarged by these cell protrusions (Fig. 8f), but unlike the pore recoil observed during transmigration, pore size remains static due to the inserted cell bodies that maintain their positions during the assay period (Fig. 8g).

**Cancer cells degrade and transmigrate basement membrane barriers, but by protease-dependent mechanisms alone**
While cancer cells are proposed to co-opt leukocyte migration programs in order to toggle between proteinase-dependent and -independent mechanisms to traverse ECM barriers, these studies have largely been confined to artificial constructs that do not recapitulate the mechanical properties of native ECM barriers[5,6,45,68,69]. As such, we next examined the ability of cancer cells to remodel and transmigrate native explants. Confirming earlier studies where human cancer cells proteolytically remodel native basement membranes[29,70], the highly invasive human breast carcinoma cell line, MDA-MB-231 or the human fibrosarcoma cell line, HT-1080, both degrade the underlying basement membrane barrier, and like macrophages, actively infiltrate the

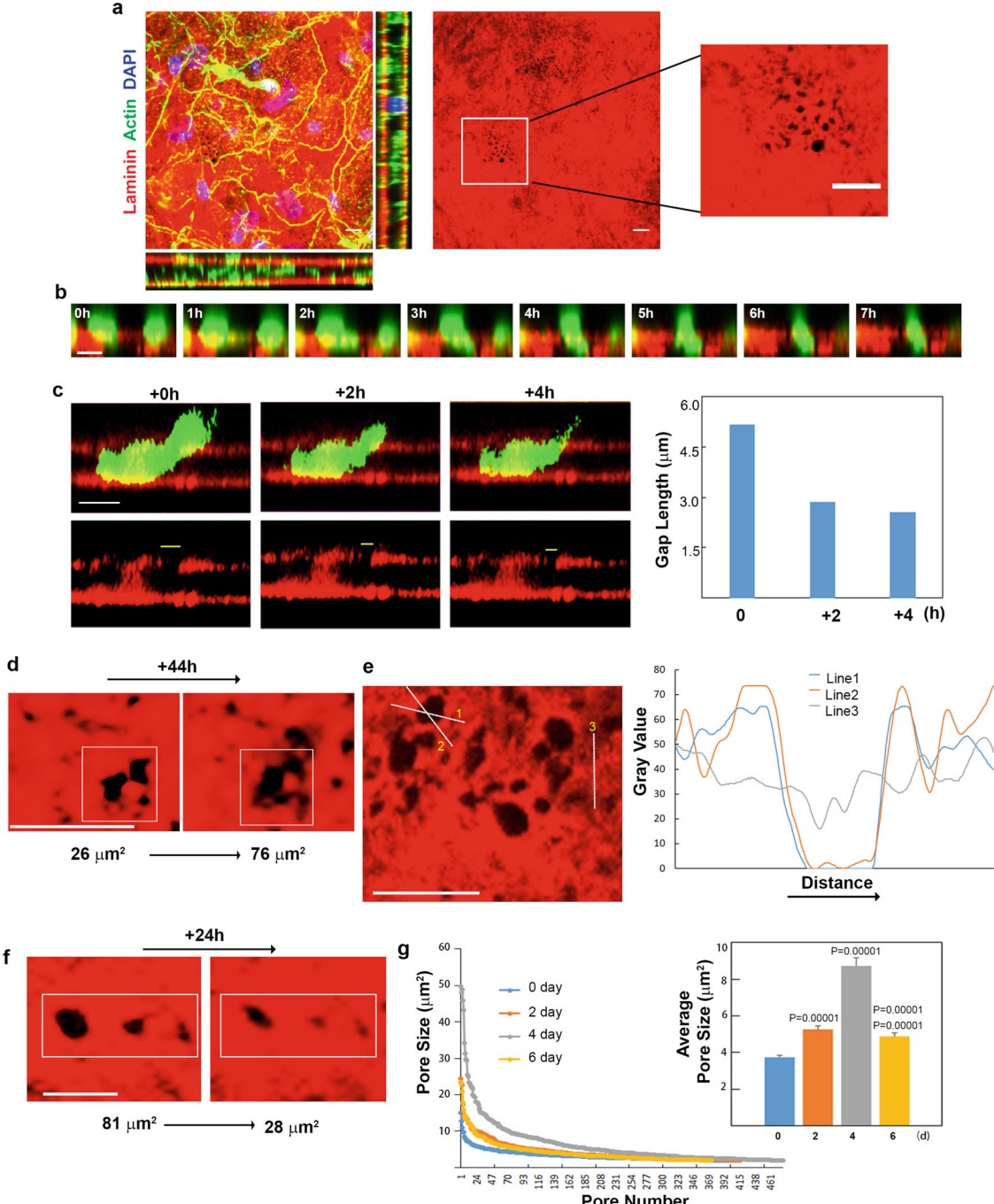

**Fig. 7 | Macrophages traverse preformed and elastic basement membrane portals. a** En face and orthogonal immunofluorescence of paraformaldehyde-fixed mesentery. In the middle panel, preformed portals are shown with the boxed region further expanded in the far right panel. Bars: 10 μm. Results representative of three independent experiments preformed. **b**, **c** Time-lapse series of CFSE-labeled human macrophages (green) and laminin-pre-labeled basement membrane (red) captured hourly for 7 h immediately after plating with BB-94 (5 μM). In **b**, LPS-polarized macrophages cultured atop a laminin-pre-labeled basement membrane (red) with BB-94 (5 μM) change cell shape while traversing preformed portals (bottom two rows). Bars: 10 μm. As the macrophage traverses a pore over 4 h (**c**, upper row), pore size decreases after the bulk of the cell navigates the entry point (lower row displays basement membrane alone with gap length quantified in the graph to the right). Results representative of three independent experiments. **d** En face view of laminin-pre-labeled basement membrane (red) cultured with LPS-polarized mouse macrophages (cells not shown for the purpose of clarity) over 44 h as pore size expands during transmigration (**d**). Results representative of three

experiments performed. **e** Normalized fluorescence intensity profiles of laminin across basement membrane perforations (lines 1 and 2) or non-transmigrated areas (line 3) in explants cultured with LPS- or treated macrophages for 2 days (**e**). Results representative of three independent experiments. Bars: 10 μm. **f** En face view of laminin-pre-labeled basement membrane (red) cultured with LPS-polarized mouse macrophages for 4 days (cells not shown for the purpose of clarity), and imaged over 24 h as pore size decreases following transmigration. Bar: 10 μm. Results representative of three independent experiments performed. **g** Size distribution of >300 basement membrane pores determined in explants cultured with LPS-polarized mouse macrophages after 0, 2, 4, and 6 days of culture. The number of pores at each size were determined using three explants for each time point. Mean pore sizes are shown in the inset. Results are expressed as mean ± SEM ($n$ = 414, 415, 480, and 373 pores quantified at 0, 2, 4, and 6 days, respectively) with significance determined by two-tailed Mann–Whitney $U$-test. Source data are provided as a Source data file.

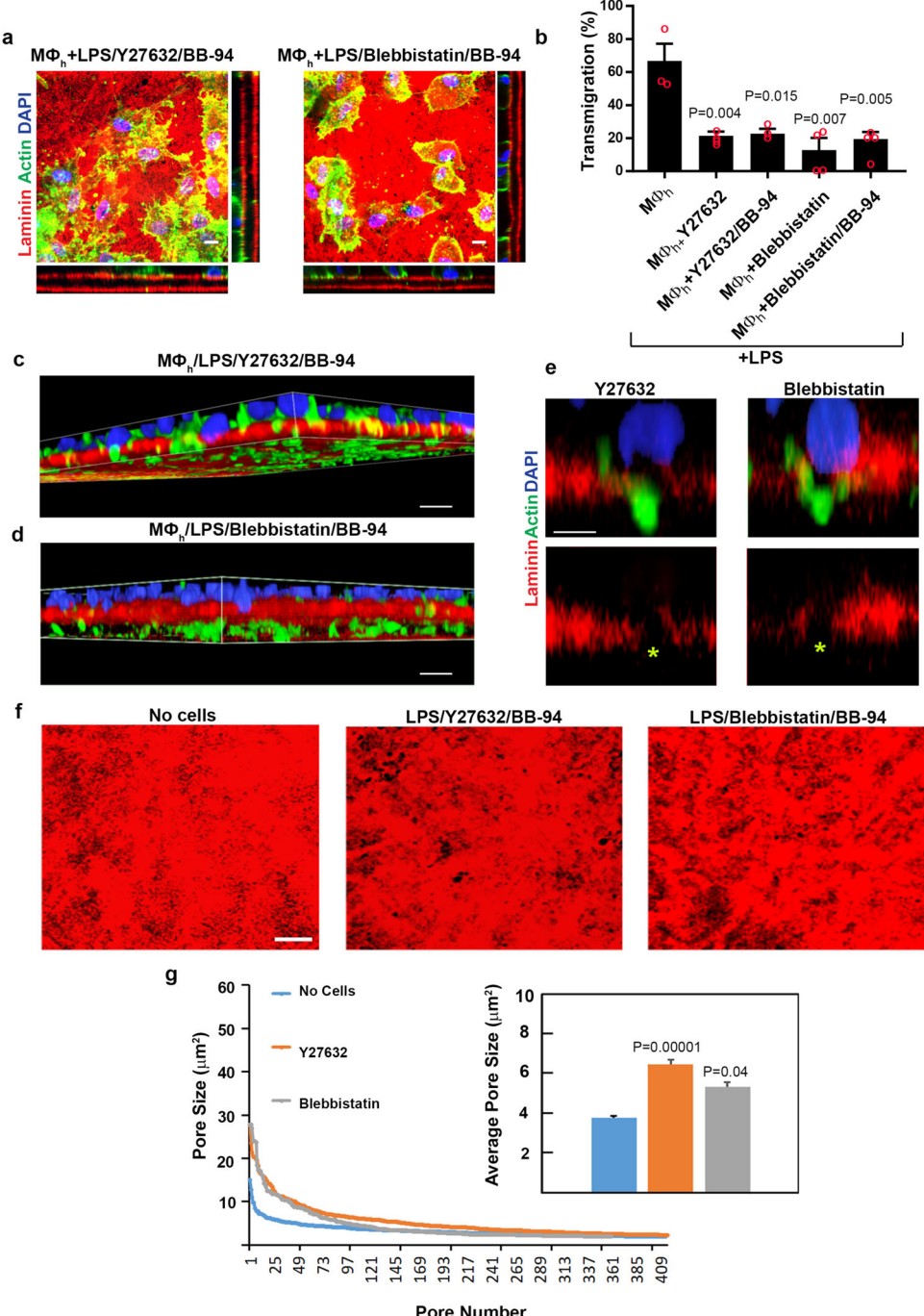

**Fig. 8 | A requirement for actomyosin-generated forces during human macrophage transmigration. a** En face and orthogonal immunofluorescence of laminin-labeled basement membrane explants cultured with human macrophages polarized with LPS (1 µg/mL) in the absence or presence of Y-27632 (20 µM) and BB-94 (5 µM) (left panels), or blebbistatin (20 µM) and BB-94 (5 µM) (right panels). Results representative of three independent experiments performed. Bars: 10 µm. **b** Quantification of transmigrated human macrophages (MØ$_h$) located between the two basement membrane sheets as a percentage of the total number of cells. Results are expressed as mean ± SEM ($n$ = 3, 4, 3, 4, and 4 independent exps, respectively for each of the ordered variables) with significance determined by two-tailed $t$ test. **c**, **d** 3D reconstructions of LPS-polarized human macrophages (F-actin labeled green with nuclei stained blue) cultured in the presence of Y27362/BB-94 (**c**) or blebbistatin/BB-94 (**d**) atop laminin-labeled basement membranes (red) for 4 days extending cell protrusions through matrix pores while nuclei remain confined to the upper surface. Results representative of 3 independent experiments

performed. Bar: 10 µm. **e** Higher-magnification image of macrophages in **c**, **d** inserting cell protrusions through basement membrane pores while macrophage nuclei remain confined to the upper surface. Bar: 5 µm. **f**, **g** En face immunofluorescent views of laminin-labeled basement membrane explants (red) cultured alone or with mouse macrophages polarized with LPS (1 µg/mL) in the absence or presence of Y-27632 (20 µM) and BB-94 (5 µM) (left panels), or blebbistatin (20 µM) and BB-94 (5 µM) showing enlarged basement membrane pores at 4 days of culture (images of macrophages have been omitted to clarify pore structure; **f**). Results representative of three independent experiments performed. Bars: 10 µm. In **g**, size distribution of >300 basement membrane pores determined in explants using 3 explants for each variable after a 4-day culture period. Mean pore sizes are shown in the inset. Results are expressed as mean ± SEM ($n$ = 300, 416, and 415 pores quantified in control versus Y-27632 or blebbistatin-treated cells in the presence of BB-94) with significance determine by two-tailed Mann–Whitney $U$-test. Source data are provided as a Source data file.

interstitial compartment (Fig. 9a–f). Unlike macrophages, however, cancer cell invasion proceeds independently of actomyosin-dependent contractility (Supplementary Fig. 4).

Recently, migrating *C. elegans* anchor cells cross basement membranes by mobilizing metalloproteinase family members[13], but in the presence of BB-94, these cells alternatively used mechanical force to physically rupture the underlying matrix, thereby allowing trans-migration to proceed independently of proteolytic remodeling[12,13]. While generally assumed that these findings are relevant to human cancer cells[6,13,71], when MDA-MB-231 or HT-1080 cells are similarly cultured atop tissue explants in the presence of BB-94, native base-ment membrane structure is maintained while cell invasive activity is largely ablated in either the absence or presence of chemotactic growth factors, though the cells do retain the ability to insert invadopodia-like structures through, presumably, basement mem-brane pores (Fig. 9a–f and Supplementary Fig. 5). Hence, in contrast to macrophages, human cancer cells are entirely reliant on MMP activity to transmigrate native tissue barriers.

While macrophages can access basement membrane pores, the apparent inability of cancer cells to follow suit may be linked to the limited malleability of their nuclei[15,22]. Indeed, recent reports have suggested that the ability of cancer cells to transmigrate small pores can be enhanced when nuclear rigidity, largely attributed to the nucleoskeletal proteins, lamin A and C, is decreased[15,22,72,73]. Despite the fact that human macrophages and cancer cells express similar levels of lamin A/C when loading is equalized for cell number (Fig. 9g and Supplementary Fig. 6), we sought to determine whether inhibiting lamin A/C expression in cancer cells might promote their tissue-invasive activity. As such, lamin A/C expression in MDA-MB-231 or HT-1080 cells was either silenced using either a series of shRNA/siRNA constructs or deleted via CRISPR/Cas9 gene editing (Supplementary Fig. 6). Whereas lamin A/C knockdown has been reported to impair MT1-MMP-dependent invasion through type I collagen hydrogels[72], lamin A/C-targeted, MDA-MB-231 or HT-1080 cells retain full degra-dative and invasive activity (Fig. 9h, i and Supplementary Fig. 6). However, in the presence of BB-94, lamin A/C-silenced MDA-MB-231 and HT-1080 cells display only limited ability to negotiate native basement membrane pores (Fig. 9h, i and Supplementary Fig. 6). Thus, in the absence of MMP activity, targeting lamin A/C expression in the human carcinoma or fibrosarcoma cell lines does not confer invasive potential, thereby highlighting the distinct processes used by macro-phages and cancer cells to remodel and traverse native tissue barriers.

**Proteinase-dependent and proteinase-independent invasion differentially regulate the macrophage transcriptome**

While macrophages are able to transmigrate the basement membrane and intervening stroma to comparable degrees via either proteinase-dependent or proteinase-independent mechanisms, proteolytic remodeling could potentially alter macrophage gene expression[74]. As such, LPS (1 μg/mL)-stimulated human macrophages were cultured in the absence or presence of BB-94 (5 μM) atop either a plastic sub-stratum or the native basement membrane for 48 h, and mRNA har-vested for transcriptional profiling. Omitting those gene changes confined to the plastic substratum alone, human macrophages engaged in the active remodeling of the basement membrane explants via proteinase-dependent versus proteinase-independent processes differentially express more than 2000 distinct transcripts (Fig. 10a). Gene ontology pathway analysis reveals that the most significant changes observed in LPS-stimulated macrophages cultured atop basement membranes in the absence or presence of MMP activity relate to inflammatory responses, defense responses and receptor activity (Fig. 10b). Of note, many of the most highly up-regulated transcripts expressed during proteolytic invasion, including CXCL8/IL-8, IL-1β, and CXCL1 (Fig. 10c), are associated with the generation of pro-inflammatory environments. By contrast, in the presence of BB-94,

non-proteolytic remodeling of the basement membrane by LPS-stimulated macrophages commits cells to a distinct transcriptional program wherein pro-inflammatory biological processes are notably absent (Fig. 10b). Instead, in the absence of MMP activity, LPS-stimulated macrophages adopt an M2-like phenotype with the highest p value assigned to "cellular responses to IL-4" (Fig. 10b). Indeed, among this gene set, MRC1, the mannose receptor C-type I, a gene product most frequently associated with the resolution of the inflammatory response[75], is upregulated more than fourfold relative to control LPS-stimulated macrophages (Fig. 10c). Hence, despite retaining similar pro-invasive activities, while matrix degradation is associated with commitment to a pro-inflammatory transcriptional program, the non-proteolytic remodeling of the ECM skews the immune response towards a phenotype more consistent with anti-inflammatory responses.

## Discussion

Macrophages as well as cancer cells participate in the remodeling of the ECM in both autonomous and cooperative fashions[3,25,76]. However, the molecular mechanisms that underlie the ability of these cells to remodel or traverse native tissue barriers, particularly basement membranes[77–81], have remained largely undefined. To date, almost all studies have relied on the use of model ECM or synthetic constructs in an effort to characterize cellular interactions with either native base-ment membrane or interstitial matrix barriers[16,17,19,20,42,43,82–86]. However, given increased appreciation that artificial constructs cannot recapi-tulate the more complex structure of the ECM in vivo, and that the composition and mechanical properties of the ECM affect cell function[21,87,88], the utility of these systems for recapitulating basement membrane remodeling in vivo is subject to debate. For example, whereas basement membranes in vivo are type IV collagen-rich and mechanically rigid as a consequence of lysyl oxidase- and peroxidasin-mediated covalent crosslinks, in vitro constructs that rely on EHS carcinoma extracts (i.e., Matrigel) are alternatively enriched with laminin, mechanically soft and largely devoid of the critical type IV collagen crosslinks that define basement membrane structure[4,5,8,89,90]. Likewise, given the fact that the interstitial matrix, though dominated by type I/III collagen, is comprised of hundreds of distinct compo-nents, attempts to faithfully mimic its structure with structurally monolithic collagen hydrogels are potentially problematic[11,45,91]. While these systems, as well as synthetic polyethylene glycol- or alginate-based substrates engineered in 2-D, 3-D, or microchannel format can provide valuable insights[22,66,87,88,92], none of these constructs authen-tically duplicate the structural complexity or architecture of native tissues. Given these limitations, we selected an explant model for characterizing cell–matrix interactions, thereby allowing us to gauge the ability of macrophages as well as cancer cells to remodel and transmigrate native basement membrane barriers while gaining access to an underlying interstitial matrix. Though our studies are confined to a single tissue, i.e., the peritoneum, we stress that (i) with regard to composition, all basement membranes are dominated by type IV col-lagen and laminin[8], (ii) type IV collagen is the single most important component in defining basement membrane mechanical integrity[93] and (iii) the peritoneal basement membrane is, in fact, transmigrated by macrophages as well as cancer cells in pathologic states[77–81].

In considering the potential proteolytic mechanisms that underlie macrophage-dependent basement membrane remodeling, we chose an unbiased transcriptional screen as a means to identify candidate proteinases. Consistent with reports implicating cysteine proteinases, serine proteinases as well as secreted MMPs in conferring macro-phages with the ability to invade Matrigel-based constructs[17,21,43,94], members of each of these proteolytic systems were expressed in polarized cells. However, when macrophage interactions with native basement membranes were examined, targeting these proteinases with class-specific inhibitors that have proven effective in model

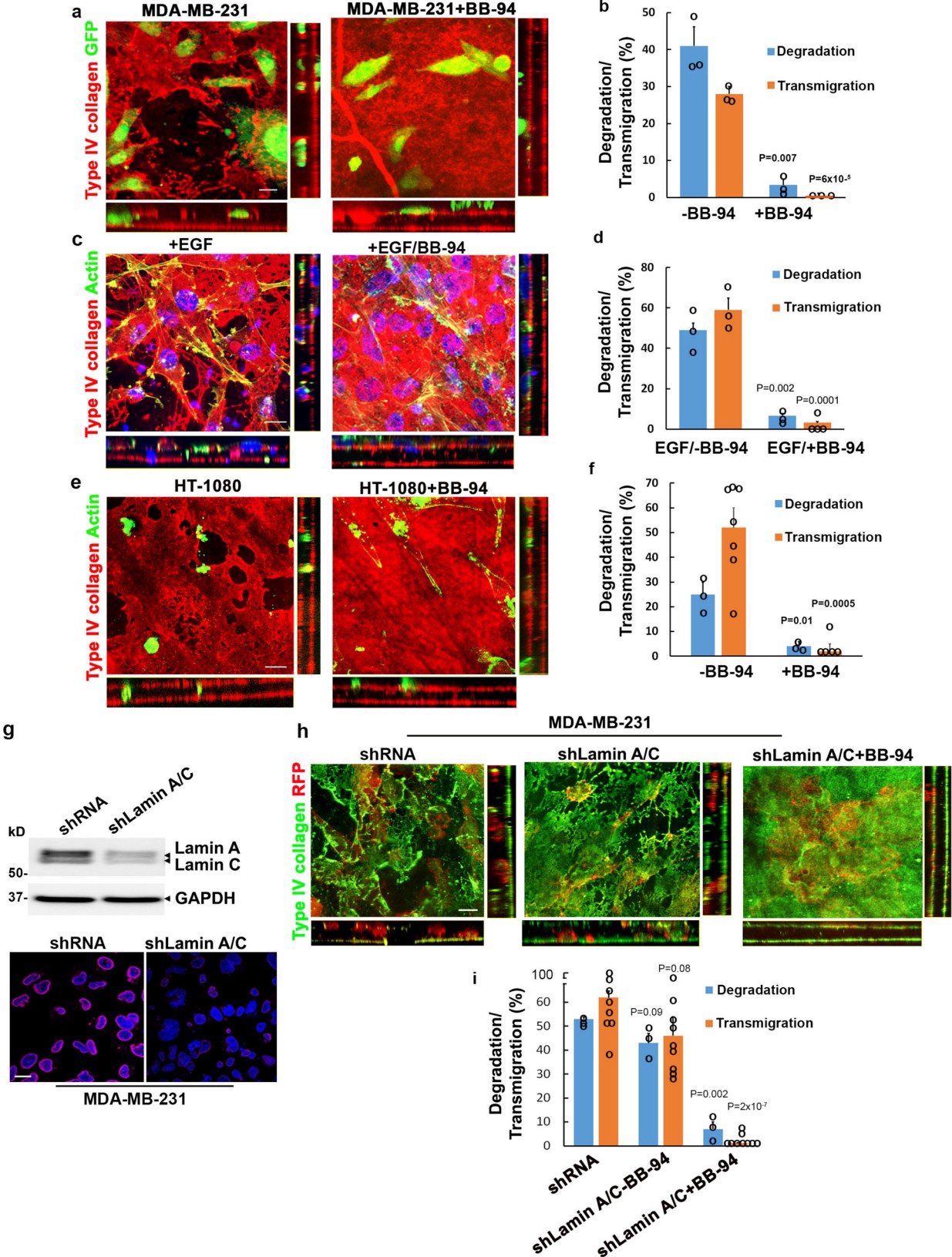

matrix systems[17,21,43,94] failed to block matrix remodeling. Instead, both human and mouse macrophages deployed the membrane-anchored MMP, MT1-MMP/Mt1-mmp, as the dominant effector of basement membrane proteolysis, a finding consistent with recent work identifying Mt1-mmp as a central regulator of basement membrane remodeling during embryonic development[95].

Following MMP targeting or inhibition, we considered the possibility that macrophages or cancer cells could mobilize cytoskeletal-generated forces to either displace or mechanically disrupt ECM fibers[6,12,13,44,96]. However, the basement membrane used here is cross-linked by covalent sulfilimine bonds at levels similar to that found in highly cross-linked tissues found in placenta[32], that would be predicted

**Fig. 9 | Divergent regulation of cancer cell invasion programs. a**, **b** En face and orthogonal images of type IV collagen-stained basement membranes cultured with $1 \times 10^5$ GFP-labeled MDA-MB-231 carcinoma cells for 4 days without or with 5 μM BB-94 (**a**). Bar: 10 μm. The percent basement membrane degraded and the percent transmigrated cells are quantified in **b**. Results are presented as the mean ± SEM ($n = 3$ independent exps) with significance determined by two-tailed $t$ test. **c**, **d** En face and orthogonal images of type IV collagen-stained basement membranes cultured with actin-labeled MDA-MB-231 carcinoma cells (green) in the presence of a chemotactic gradient of EGF (10 ng/mL) for 4 days without or with 5 μM BB-94 (**c**). Bar: 10 μm. The percent basement membrane degraded and the percent transmigrated cells are quantified in **d**. Results are presented as the mean ± SEM ($n = 3$ independent exps) with significance determined by two-tailed $t$ test. **e**, **f** En face and orthogonal images of type IV collagen-stained basement membranes cultured with $1 \times 10^5$ actin-labeled HT-1080 fibrosarcoma cells (green) for 4 days without or with 5 μM BB-94 (**e**). Bar: 10 μm. The percent basement membrane degraded and the percent transmigrated cells are quantified in **f**. Results are presented as the mean ± SEM ($n = 3$ independent exps for degradation with $n = 7$ and 5, respectively, for transmigration in the absence or presence of BB-94) with significance

determined by two-tailed $t$ test. **g** Lamin A/C expression was silenced with a specific shRNA construct and the cells either lysed for immunoblotting or immunostained for lamin A/C expression (stained red with DAPI counterstained blue). Results using a second siRNA construct are shown in Supplementary Fig. 5. Bar: 10 μm. All experiments are representative of three experiments performed. **h** RFP-labeled MDA-MB-231 cells transduced with an shRNA control or shLamin A/C were cultured atop basement membrane in the absence or presence of BB-94 explants for 5 days, immunostained for type IV collagen and imaged for en face and orthogonal views. Results representative of three independent experiments performed. Bar = 10 μm. **i** The percent basement membrane degraded and percent transmigrated MDA-MB-231 cells in panel h was quantified. Results are expressed as the mean ± SEM of 3 independent experiments for degradation and 8, 9, and 9 independent experiments performed for transmigration of shRNA control-treated, shLamin A/C-treated, and shLamin A/C-treated cells cultured in the presence of BB-94, respectively, with significance of shRNA control-treated cells versus shLamin A/C-treated cells cultured without BB-94, and shLamin A/C-treated cells cultured without BB-94 versus with BB-94 determined by two-tailed $t$ test. Source data are provided as a Source data file.

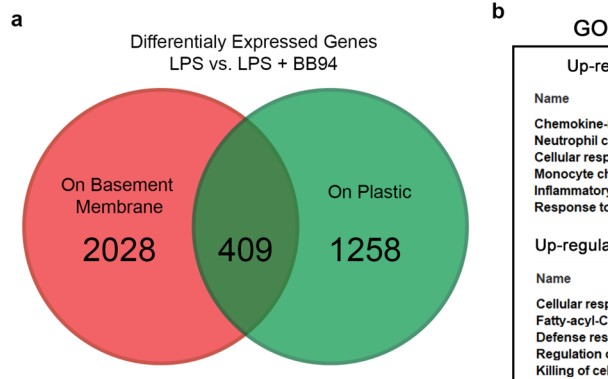

**a**

Differentialy Expressed Genes
LPS vs. LPS + BB94

On Basement Membrane: 2028 | 409 | On Plastic: 1258

**b**

### GO Terms - Biological Processes

Up-regulated in $M\Phi_{LPS}$ / Basement Membrane

| Name | # genes (DE/ALL) | p-value |
|---|---|---|
| Chemokine-mediated signaling pathway | (17 / 74) | 9.00E-08 |
| Neutrophil chemotaxis | (20 / 85) | 1.20E-07 |
| Cellular response to interleukin-1 | (17 / 86) | 1.80E-07 |
| Monocyte chemotaxis | (12 / 55) | 3.90E-06 |
| Inflammatory response | (72 / 638) | 1.00E-05 |
| Response to lipopolysaccharide | (37 / 295) | 3.70E-05 |

Up-regulated in $M\Phi_{LPS}$ + BB94 / Basement Membrane

| Name | # genes (DE/ALL) | p-value |
|---|---|---|
| Cellular response to interleukin-4 | (8 / 29) | 4.70E-05 |
| Fatty-acyl-CoA biosynthetic process | (7 / 29) | 2.00E-03 |
| Defense response to Gram-positive bacterium | (10 / 74) | 3.00E-03 |
| Regulation of receptor activity | (38 / 503) | 4.00E-03 |
| Killing of cells of another organism | (7 / 48) | 8.00E-03 |
| Oxidation-reduction process | (60 / 915) | 8.00E-03 |

**c**

### Top Up-Regulated Genes

| LPS | | LPS+BB-94 | |
|---|---|---|---|
| Gene | Fold-Change | Gene | Fold-Change |
| EREG | 4.94 | MRC1 | 4.43 |
| CCL23 | 3.17 | MMP12 | 3.74 |
| CXCL8 | 2.90 | CIITA | 2.88 |
| ANKRD1 | 2.85 | HAMP | 2.61 |
| CD70 | 2.62 | CAMK2D | 2.50 |
| JAK3 | 2.44 | CISH | 2.43 |
| IFNA10 | 2.44 | TNFSF13 | 2.38 |
| IL2RA | 2.35 | MIR31 | 2.37 |
| CXCL3 | 2.24 | FABP4 | 2.35 |
| CCL1 | 2.23 | TNFSF8 | 2.30 |
| IL1β | 2.15 | CCL17 | 2.25 |
| SERPINB2 | 2.08 | MME | 2.24 |
| CXCL1 | 2.07 | ACKR4 | 2.22 |
| CCL8 | 1.99 | CXCR1 | 2.22 |
| GFPT2 | 1.98 | TREM2 | 2.22 |
| MT1X | 1.97 | HIST1H3H | 2.17 |
| FYN | 1.94 | TFRC | 2.08 |
| STAT4 | 1.92 | CSF3 | 2.06 |
| CD40LG | 1.90 | SPRY2 | 2.03 |
| CCL14 | 1.90 | NYX | 2.02 |

**Fig. 10 | MMP-dependent regulation of macrophage-basement membrane transcriptional responses. a** LPS-stimulated human macrophages were cultured atop a standard tissue culture plastic substratum or basement membrane explant for 48 h in the absence or presence of BB-94 (5 μM) and transcriptional responses quantified in 2 independent experiments. Differences in gene expression (1.5-fold enrichment cutoff) detected on plastic surfaces versus atop basement membrane

explants are presented in green and red, respectively. Results representative of two independent experiments performed. **b** Gene ontology pathway analysis of biological responses differentially affected in the absence or presence of BB-94. **c** The top 20 upregulated and downregulated transcripts and fold changes are listed for macrophages cultured atop basement membrane explants in the absence or presence of BB-94. Source data are provided as a Source data file.

to render the type IV collagen backbone resistant to mechanical displacement[97]. Further, while migrating cells can negotiate fixed pores whose size exceeds 10% of the nuclear cross-sectional area, the type IV collagen network is estimated to limit interfibrillar pore size to ~50 nm in diameter, dimensions that would be predicted to effectively preclude cellular transmigration[10,45,98,99]. However, we find that both human and mouse macrophages retain the ability to cross basement membrane-interstitial matrix interfaces via an MMP-independent process that requires actomyosin-generated forces. In our efforts to visualize the sites permissive for proteinase-independent transmigration, our attention focused on discrete ~3 μm² pores that decorate the basement membrane surface. Importantly, micrometer-sized basement pores have been identified in lung, skin, blood vessel, and colon tissues, supporting the proposition that these structures are generated purposefully during embryogenesis to allow epithelial/mesodermal-stromal crosstalk as well as serve as permissive passageways for normal cell trafficking[44,59,60,63,64]. Given their relatively small pore size—at least relative to the nuclear dimensions of most cell populations—the engagement of the macrophage actomyosin network is consistent with recent studies demonstrating similar cytoskeletal requirements for modulating cell as well as nuclear shape when negotiating space-restrictive environments[22,66,92]. Interestingly, although the basement membrane surrounding lymphatic vessels are less well-organized than those found subtending epithelial cells or the vascular endothelium, Sixt and colleagues reported that basement membrane portals similar to those described here are permissive for non-proteolytic dendritic cell trafficking[44]. Further, as observed in our work, the basement membrane displayed elastic qualities that allowed expanded pores to retract following cell transmigration[44]. Likewise, in studies that complement our findings, van den Berg et al used a zebrafish model to demonstrate that macrophages can breach basement membranes in vivo by either mobilizing proteolytic enzymes or alternatively using a protease-independent mechanism to preferentially move through pre-existing "weak spots" or pores in the basement membrane[100].

Like macrophages, cancer cells are also able to degrade and penetrate the basement membrane, but invasion was inhibited to near background levels when MMP activity was targeted in peritoneal as well as human dermal explants, in ovo in the chick chorioallantoic membrane, and similarly in a mouse transgenic model of breast cancer in vivo[4,22,29,55,70]. Nevertheless, recent reports have emphasized the ability of cancer cells, including MDA-MB-231 carcinoma cells or HT-1080 fibrosarcoma cells, to alternatively adopt an ameboid phenotype to negotiate model matrix barriers via non-proteolytic mechanisms[6,23,87,101–103]. Based on our studies, and in contrast to macrophages, we conclude that cancer cells are largely dependent on MMP-dependent proteolysis to traverse native tissue barriers[11,29,55,70]. Interestingly, independent of its proteolytic activity, MT1-MMP has been reported to generate pushing forces at cancer cell–ECM contacts, indicating that the protease can coordinate proteolysis with mechanical activity[104]. However, as this MT1-MMP-dependent, force producing activity is unaffected by MMP inhibitors[104], our results demonstrate that this pushing force alone is incapable of driving basement membrane remodeling.

Notably, these results stand in direct contrast to recent studies wherein anchor cell invasion into vulval tissues during *C. elegans* development was proposed as a model that helped shed light on the inability of MMP inhibitors to block cancer cell invasion in human patients[6,12,13]. Using this model organism, the authors identified a MMP-independent process that allowed anchor cells to physically breach the basement membrane by exerting F-actin-based mechanical forces[12,13]. However, caution should be exercised in extrapolating results from developmental anchor cell invasion to human cancer cells transmigrating native connective tissue barriers. First, the mechanical properties and status of type IV collagen crosslinks in the developing worm remain undefined. Second, while proofs of MMP-independent cancer

invasion are based frequently on the ineffectiveness of MMP inhibitors in clinical trials[6,13], the peak plasma concentrations of MMP inhibitors used in these studies fall far below those needed to inhibit membrane-anchored MMPs[11]. Third, the detection of proteinase-independent invasion programs in vivo are uniformly limited to studies that employ short-term observation periods to accommodate real-time imaging, thereby circumventing efforts to track cancer cell invasion over extended time periods as the cell encounters multiple ECM barriers[5,11,55]. As shown here, in the absence of MMP activity—unlike anchor cell invasion—carcinoma cells exhibited little if any ability to remodel or breach a functionally relevant basement membrane barrier, and were unable to mount an effective invasion program. Though nuclear lamin A/C-dependent stiffness has been reported to serve as a barrier for 3D migration[6,14,15], targeting lamin A/C expression in HT-1080 or MDA-MB-231 cancer cells also did not prove permissive for proteinase-independent invasion, despite their full retention of proteolytic activity. While we have been unable to identify a proteinase-independent cancer cell invasion program in our model, Glentis et al have proposed that cancer-associated fibroblasts can promote cancer cell invasion through basement membrane explants similar to ours by a metalloproteinase-independent process[105]. However, in their model, fibroblasts were applied to the face of one of the basement membrane surfaces in a fashion that does not recapitulate normal tissue architecture where stromal cells reside solely within the interstitial matrix. Hence, the ability of stromal fibroblasts to provide cancer cells with protease-independent access to basement membrane pores remains possible, but the conclusions reached by Glentis et al.[105] require further study.

Finally, given the ability of macrophages to traverse basement membrane barriers in either the absence or presence of proteolytic remodeling, we sought to determine whether the mode of transmigration might potentially impact their selection of transcriptional programs. While macrophage invasion proceeds comparably in the absence or presence of proteinase inhibitors, more than 2000 unique gene products are differentially affected in a basement membrane-specific fashion, with the bulk of the affected transcripts consigned to inflammatory/immune-related pathways. As proteolytic remodeling can potentially trigger changes in gene expression by altering macrophage adhesion and cell shape via the release of matrix-bound growth factors, the generation of bioactive matrix degradation products or changing ECM rigidity, efforts to deconvolute this complex process will prove difficult as these changes occur in a coupled fashion. Nevertheless, these studies highlight the fact that the deployment of proteinase-dependent versus proteinase-independent pathways at the macrophage–ECM interface exerts substantive effects on skewing transcriptional responses to either pro-inflammatory or reparative states. These results potentially align with recent descriptions of intraepithelial macrophages that have crossed the normal mammary gland basement membrane[106] or even the presence of macrophage protrusions penetrating the epithelial basement membrane in normal colon[107] or the vascular basement membrane in skin[57]. In each of these cases, given the normal structure of these basement membranes, we posit that macrophages traverse these barriers by accessing resident pores without engaging proteolytic activity while maintaining a quiescent (i.e., non-inflammatory) phenotype. In this regard, recent studies in *Drosophila* have shown that Mmp2-dependent basement membrane damage induces a pro-inflammatory response[108].

In sum, we find that macrophages mobilize MT1-MMP as the dominant proteolytic effector of basement membrane remodeling during transmigration. Cancer cells can also deploy MT1-MMP, as well as membrane-anchored MMPs, MT2-MMP and MT3-MMP, to remodel basement membranes during trafficking[29,70]. However, though the underlying mechanism remain to be defined, only macrophages are able to adopt an alternate tissue-invasive phenotype that allows them

to transmigrate both basement membrane and stromal tissue barriers in a proteinase-independent fashion. Importantly, this dual ability allows macrophages to modulate transcriptional responses when infiltrating tissues and proteolytically remodeling the ECM in association with tissue-destructive events versus using mechanical forces in order to purposefully leave tissues unscathed during reparative states.

## Methods

All research performed herein complies with all relevant ethical regulations and has been approved by the University of Michigan Institutional Review Board and the Institutional Animal Care & Use Committee.

### Isolation of primary macrophages

Bone marrow macrophages were isolated from both male and female 2–8-week-old wild-type (*Mt1-mmp*[+/+]) or Mt1-mmp-null (*Mt1-mmp*[−/−]) Swiss Black mice (animal use approval code PRO00010618) housed under standard conditions (12 h light/12 h dark cycle at 22 °C and 30–40% humidity)[109]. Long bones were flushed with PBS, red cells were lysed with ACK buffer (Thermo Fisher) and the remaining cells were cultured in alpha-MEM with 10% heat-inactivated fetal bovine serum (HI-FBS), 1% penicillin–streptomycin solution (Thermo Fisher), and 10 ng/mL M-CSF (R&D Systems) overnight on tissue culture dishes. Non-adherent cells were plated onto non-tissue culture-treated dishes in media with M-CSF for an additional 5–7 days; media was replaced every 48 h. Human peripheral blood monocytes were isolated from whole blood of male and female volunteers in accordance with the University of Michigan Institutional Review Board (IRB) approval and volunteer donor pool participants' informed consent (IRBMED# 1987-0242). PBMCs were separated by Lymphocyte Separation Medium (Corning) by density centrifugation, purified by CD14 selection (Miltenyi Biotec) and cultured at $2 \times 10^6$ in 6-well plates containing RPMI 1640 without serum. After 2 h, media was replaced with RPMI 1640 with 1% penicillin-streptomycin solution and 20% autologous serum for 5–7 days. Autologous serum was prepared by incubating non-heparinized whole blood at 37 °C for 1 h followed by centrifugation at $2850 \times g$ for 15 min, and sterile filtration of the serum fraction.

### Ex vivo mesentery ECM preparation

Following surgical removal from exsanguinated male or female outbred rats (approval code PRO00009587), mesentery explants were mounted on 6.5- or 12-mm diameter Transwells (Sigma) with sterile surgical thread and decellularized with 0.1 N ammonium hydroxide[29]. In brief, rat mesentery. $1 \times 10^5$ mouse or human macrophages were cultured atop the tissue for 6 days with media changes every 48 h. All experiments were performed in complete medium in the absence or presence of the following inhibitors 100 μM E-64d, 100 μg/mL aprotinin, 100 μg/mL soybean trypsin inhibitor (SBTI), 20 μM Y-27632, 20 μM blebbistatin (Sigma), 5 μM BB-94 (Tocris Bioscience) or 12.5 μg/mL TIMP-1, 5 μg/mL TIMP-2 (Peprotech). Protease inhibitor mix contained 100 μM E-64d, 100 μg/mL aprotinin, 100 μg/mL SBTI, 5 μM BB-94, 2 μM leupeptin, 10 μM pepstatin A[45]. Human macrophages were also cultured with either 75 μg/mL human isotype control IgG antibody or anti-MT1-MMP antibody DX-2400[53] in medium with 20% heat-inactivated autologous human serum and 1% penicillin-streptomycin in the presence of 5 μL $F_c$-receptor blocking antibody TruStain FcX (Biolegend). DX-2400 was provided by the Kadmon Corporation. Macrophages were polarized with 1 μg/mL LPS from *Escherichia coli* O111:B4 (Sigma) or 20 ng/mL recombinant mouse or human IL-4 (Peprotech). After 6 days of culture, tissue constructs were washed with PBS, fixed with 4% PFA, and stained as described.

### Lentiviral gene transfer

A mCherry-tagged MT1-MMP construct[52] was cloned into pLenti lox IRES EGFP vector and subsequently transfected into 293 T cells using Lipofectamine 2000 (ThermoFisher) to generate lentiviral particles. Mouse bone marrow-derived macrophages at 5 days post-isolation were incubated with the lentivirus-containing supernatant in the presence of 8 μg/mL polybrene for 6 h before media was replaced. Forty-eight hours later, transduced macrophages were cultured atop the tissue construct as described.

### Tumor cell culture

In all, $1 \times 10^5$ MDA-MB-231 breast carcinoma (ATCC) or HT-1080 fibrosarcoma cells (ATCC) were cultured in DMEM supplemented with 10% heat-inactivated fetal bovine serum (HI-FBS) and a 1% penicillin–streptomycin solution. In selected experiments, cancer cells were transfected with the invadopodial marker, Tks5-GFP as described[104]. Cells were cultured on the basement membrane construct for 4–5 days before processing.

### Confocal fluorescence microscopy and analysis

PFA-fixed constructs were incubated with polyclonal antibodies targeting laminin (Sigma, cat #: L9393), type IV collagen (Abcam, cat #: ab19808) or elastin (EMD Millipore; cat # 2039) at 1:150 dilution in a blocking solution of 1% bovine serum albumin-PBS for 1 h room temperature. Constructs were then incubated with secondary fluorescent antibodies at 1:250 dilution while cells were labeled with Alexa Fluor 488 phalloidin and DAPI (Sigma) for 1 h in blocking solution. Image acquisition was performed using a spinning disc confocal CSU-WI (Yokogawa) on a Nikon Eclipse TI inverted microscope with a ×60 oil-immersion objective and the Micro-Manager software (Open Imaging) or using Nikon NIS-Elements AR (v. 5.11.03). Fluorescent images were processed with ImageJ (National Institutes of Health) with 3D viewer plugin for orthogonal and 3D reconstructions. Confocal imaging of the collagen I matrix was captured by second harmonic generation on a Leica SP5 inverted confocal microscope with a ×60 oil-immersion objective.

For immunofluorescence of endogenous MT1-MMP, fixed primary human macrophages were incubated on glass coverslips with 1:50 rabbit monoclonal anti-MT1-MMP (Abcam) overnight at 4 °C in 3% BSA-PBS with or without 0.1% Triton X-100 to permeabilize the cells, followed by incubation with 1:200 Alexa Fluor-488 secondary antibody for 1 h 37 °C. For lamin A/C immunostaining, cells were fixed with 4% FPA and permeabilized with 1% Triton in PBS for 30 min prior to staining with lamin A/C rabbit antibody (clone 4C11, Cell Signaling). Additional antibody information is available in Supplementary Table 1.

### Live image microscopy

Live imaging was performed on unfixed tissue constructs pre-labeled with fluorescent antibodies as above. Macrophages were incubated with 5 μM CFSE (Life Technologies) in PBS for 20 min at 37 °C, quenched with a 5× volume of medium with 1% HI-autologous serum, resuspended in PBS-$F_c$ receptor block for 5 min at room temperature, and plated on the pre-labeled tissue construct. Z-stacks or single slices were captured in a 37 °C 5% $CO_2$ humidified chamber (Livecell Pathology Devices). Cell outlines were generated and overlaid using the binary and outline functions of ImageJ.

### Electron microscopy

Tissue constructs were processed for SEM as follows, fix in 2% glutaraldehyde/1.5% paraformaldehyde in 0.1 M cacodylate buffer, post-fix in 1% osmium tetroxide, and dehydrated through a graded ethanol series as described[29]. Image acquisition was performed using an AMRAY 1910 field emission scanning electron microscope at 5.0 kV.

### ELISA

Anti-rat COL4A1 ELISA kits were purchased from LSBio. Tissue constructs were cultured for 72 h in the presence of media alone or with

LPS-stimulated human macrophages and LPS before cell-free media was analyzed according to the manufacturer's instructions.

### qPCR and transcriptional profiling

RNA was isolated from macrophages using the NucleoSpin RNA kit (Macherey Nagel) as instructed. Day 5–7 macrophages were polarized for 24 h as described. cDNA synthesis was performed with Superscript III enzyme (Invitrogen). qPCR reactions were performed in triplicate with SYBR green PCR master mix on a 7900HT fast Real-Time PCR machine (Applied Biosystems). Data were analyzed using the comparative threshold cycle method with mRNA levels normalized to GAPDH. Primer sequences are available in Supplementary Table 2.

For transcriptional profiling, total mRNA was isolated as above, and labeled and hybridized to Mouse Gene ST 2.1 strips (Affymetrix). Three replicates of each sample were analyzed by the University of Michigan Microarray Core. For mouse macrophage proteases expression analyses, values >$2^4$ in any condition were tabulated and further analyzed for relative fold differences across conditions. For human macrophage gene expression analyses, expression values for each probe set were calculated using a robust multi-array average (RMA)[110] and filtered for genes with a greater than 1.5-fold change.

### Western blot

Western blots were performed as described with antibodies targeting MT1-MMP (Epitomics), alpha 2 (IV) NC1, clone H22 (Chondrex), lamin A/C, clone 4C11 (Cell Signaling), β-actin or GAPDH (both from Cell Signaling). Primary antibodies were labeled with horseradish peroxidase-conjugated species-specific secondary antibodies (Santa Cruz) and detected by the SuperSignal West Pico system (Pierce). For type IV collagen dimer-monomer content analysis, isolated tissue was first digested with bacterial collagenase type IV (Worthington Biochemical Corporation) overnight at 37 °C with occasional vortexing. Samples were pelleted at $15,000 \times g$ for 20 min and analyzed by SDS-PAGE without heat-denaturation. Uncropped and unprocessed images of all gels are available in the Source Data files. Additional antibody information is available in Supplementary Table 1.

### Lamin A/C targeting

Lamin A/C siRNA (Santa Cruz) was introduced into cells using Pepmute siRNA transfection reagent (Signagen) per the manufacturer's guidelines. Lamin A/C-specific TRIPZ lentiviral shRNA was obtained from Horizon (RHS5087-EG4000). Transduced cells were puromycin selected (5 μg/mL), and induced with doxycycline (1 μg/mL). For gene editing, plasmid pX330-U6-Chimeric_BB-CBh-hSpCas9 (gift from F. Zhang; Addgene plasmid # 42230; http://n2t.net/addgene:42230; RRID: Addgene_42230). We modified the plasmid by introducing a copy of a CMV promoter-controlled RFP gene into the construct. Two lamin A/C-specific sgRNA: sg1: TCTCAGTGAGAAGCGCACGCTGG and sg2: GGCGAGCTGCATGATCTGCGGGG were cloned into the modified plasmid. Lamin A/C KO cells were selected by RFP flow sorting.

### Statistical analysis

The area of basement membrane degraded as well as basement membrane perforation size were calculated using ImageJ as follows; image intensity was adjusted and background subtracted using default settings, the images converted to black and white via binary function, inverted, and particles larger than a 1 μm$^2$ minimum cut-off analyzed. For percent invasion, cells were considered traversed if the cell body, including nucleus, were located between the two basement membrane layers. Results are expressed as mean ± SEM with statistical analysis performed using an unpaired two-tailed Student's $t$ test. For changes in basement membrane pore size, the two-tailed Mann−Whitney $U$-test was used. For all tests, $p \le 0.01$ was considered statistically significant.

### Reporting summary

Further information on research design is available in the Nature Research Reporting Summary linked to this article.

## Data availability

The Affymetrix microarray data have been deposited in the NCBI Gene Expression Omnibus (GEO) database and are available through accession numbers: GSE122823 (array of *Mt1-mmp*$^{+/+}$ and *Mt1-mmp*$^{-/-}$ mouse macrophages in response to LPS or IL-4), GSE122824 (array of human macrophages on plastic or mesentery tissue in the presence or absence of BB-94), and GSE122825 (array of *Mt1-mmp*$^{+/+}$ and *Mt1-mmp*$^{-/-}$ mouse macrophages and human macrophages in the presence or absence of MMP inhibitors). All data used in this manuscript have been made available in the article, Supplementary Information, and accompanying Source data files, and in the repositories listed above. Full image datasets can be made available upon request. Source data are provided with this paper.

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

## Acknowledgements
We thank the Kadmon Corporation for providing the anti-MT1-MMP antibody, licensed to the Kadmon Corporation by the Dyax Corporation. We acknowledge Stephanie King (Life Sciences Institute) for assistance with illustrations and Craig Johnson (University of Michigan) for assistance with microarray analysis. We thank Stefan Linder for helpful discussions. This work was supported by grants from the NIH to SJW (AI105068 and AR065524), the Breast Cancer Research Foundation (S.J.W), Margolies Family Discovery Fund for Cancer Research, and the Cancer Biology Training Grant (NCI Training Grant T32-CA009676).

## Author contributions
J.C.B and X.-Y.L. planned and performed experiments and participated in writing the manuscript; T.Y.F. performed analyses; L.J. performed experiments; and S.J.W. oversaw the project, planned experiments, and wrote the manuscript.

## Competing interests
The authors declare no competing interests.
