## [Peer Review File · Nature Communications]

Divergent Regulation of Basement Membrane Trafficking by Human Macrophages and Cancer CellsREVIEWER COMMENTS

Reviewer #1 (Remarks to the Author): with expertise in cancer cell migration and invasion

Macrophages are known to infiltrate interstitial tissues during host defense mechanisms and inflammation. In the present work, using a decellularized mesentery extracellular matrix (ECM) tissue system that is mastered by this group, the authors addressed the mechanisms of basement membrane (BM) remodeling by mouse and human primary macrophages. Similar to previous work from this lab using several cellular systems including cancer-derived cell lines, the authors report that macrophages are able to remodel the BM in a MT1-MMP/MMP14-dependent manner. In addition, they found that LPS stimulation induces the expression of several proteinases in polarized macrophages, which correlates with a robust increase in their BM-remodeling capacity. However, despite mounting a strong proteinase response, polarized macrophages migrate across the BM in a proteinase- and MT1-MMP-independent and actomyosin-dependent manner, in sharp contrast with cancer cells that rely strictly on the MT1-MMP-dependent program for BM transmigration. Finally, MMP-dependent remodeling of the BM by LPS-stimulated macrophages is shown to induce a full transcriptional program possibly linking ECM remodeling and MT1-MMP function to host defense and inflammatory responses.

Overall, results are very neat and clear-cut, however some data regarding the dependency of the BM remodeling program on MT1-MMP are largely confirmatory of previous findings and long-established knowledge by this group based on different cellular models. In addition, some issues should be addressed in order to strengthen the manuscript, which are listed below (in the order of the figures).

- 1- Based on data such as in Figure 1I, it seems that macrophages opportunistically make use of and widen preexisting pores in ex vivo mesentery BM, raising the issue of whether macrophages can actually pierce an intact BM and whether the observations are relevant to the in vivo situation.
- 2- Perforation of the basal BM in mesentery tissue seems to be a rare event based on the images provided in Figure 2B. If relevant, these events should be carefully quantified.
- 3- Along similar lines, the authors state that macrophages were seen to encircle some apical BM perforations with no sign of protrusion or to send small protrusions through the cavity. Based on Video 5 and Video 6, again these seem to be rare events and they should be quantified. In addition, matrix distortion events and evidence for local force production are not so clear from the movies and these should be highlighted and quantified.
- 4- Based on the immunoblotting analysis provided in Figure 5A, MT1-MMP levels are down-modulated in IL-4 treated macrophages. Yet, BM degradation of IL-4 treated vs control macrophages is similar (Figure 2E), raising the issue of the implication of MT1-MMP in BM remodeling by IL-4 polarized cells.
- 5- While both macrophages and breast tumor cells require MT1-MMP for BM remodeling, data in figure 7 show that these cell types differ in their requirement for MT1-MMP for transmigration through the BM; i.e. migration of macrophages through the BM is MT1-MMP independent in contrast to the strict requirement of tumor MDA-MB-231 and HT1080 cells for MT1-MMP. In addition, the migration of macrophages through the BM is nicely shown to require actomyosin contractility. These are rather preliminary observations that should be strengthened by analyzing the effect of blebbistatin/Y27632 in more detail for example looking at the effect of these compounds on earlier described phenotypes such as perforation encirclement and protrusion formation using live cell imaging. It would also be interesting to test the effect of actomyosin inhibitors on the transmigration of cancer cells through BM in the mesentery tissue model system in order to assess the generality or specificity of the actomyosin dependency.

Reviewer #2 (Remarks to the Author): with expertise in cancer cell migration and invasion

Bahr et al. present a study investigating protease dependent and independent interactions between human/mouse macrophages or HT-1080/MDA-MB231 cancer cells with native matrix explants. They conclude that macrophages and cancer cells rely on MT1-MMP for basement membrane remodeling

and transmigration. However, they find that while macrophages are able to transmigrate across the basement membrane in the presence of protease inhibitors, cancer cells wholly depend on protease activity for transmigration.

This manuscript presents some compelling and important ideas and results utilizing a powerful model system, however, major revisions are required for this manuscript to become suitable for publication in Nature Communications. The model system of basement membrane invasion is quite powerful, and the finding that macrophages can traffic through basement membranes independent of proteases is an important new finding. The potential finding of protease-dependent invasion required for invasion of cancer cells through rat mesentery BM is important and adds to the current literature in this area. There are also some interesting experiments looking at macrophage activation and phenotype and the impact on remodeling. However, there are some major issues with this manuscript that need to be addressed. Many of the claims presented in this manuscript are vastly overstated and need to be moderated. Much of the analysis is qualitative and conclusions based on representative images; additional rigorous quantitative analysis is required for many of the results. Finally, some additional mechanistic insight into pore dynamics during macrophage invasion would make this stronger.

Major comments:

1. Some of the claims made in this manuscript are far too broad, and either need to be moderated, or supplemented with additional experiments or analyses that better justify their broad claims.
 - a. Extrapolation of claims beyond mesentery basement membranes: In a number of places, the authors generalize their results to make conclusions about basement membrane remodeling and invasion very broadly. However, they have done studies in only one basement membrane system – rat mesentery basement membrane. It is well known that the composition (e.g. type of laminin), architecture (e.g. thickness), and mechanical properties of basement membrane vary widely in humans. Rat mesentery basement membrane is not representative of all human BMs, and I don't believe that mesentery BM is even traversed in most cancers. Further, the BM is prepared in ways that could potentially impact the structure of the rat mesentery BM. Thus, the claims like "By contrast, cancer cell invasive activity is completely reliant on metalloproteinase activity and neither mechanical force nor changes in nuclear rigidity rescue transmigration" or "Hence, in contrast to macrophages, human cancer cells are entirely reliant on MMP activity to transmigrate native tissue barriers", and similar statements, need to be qualified, and the fact that rat mesentery BM is studied here should be specified in the abstract. If the authors wanted to make broader claims, they would have to show these results in other physiological models of BM (for example BM of vasculature might be possible).
 - b. Claim about protease dependence of cancer cell invasion: In this assay, cancer cells could simply continue migrating along the plane of the BM, where migration is unimpeded (in contrast to native tissue environment around a BM), as migrating cells often follow paths of least resistance. The cells don't necessarily have to invade through the BM if they are not driven/forced to (for example by a strong chemokine gradient). Thus, the lack of invasion of cancer cells through the BM, doesn't mean that it is not possible – it may be that the cell is just not activated sufficiently to do so. This is why one has to be careful making such strong definitive statements that cancer cells cannot invade through BM at all, independent of proteases.
 - c. Even if the authors are able to more strongly support their claim in b, these data are not inconsistent with the findings of Glentis, et al., that CAFs open up pores for cancer cells to invade through. That is still a distinct possibility and it should be acknowledged that cancer cell invasive activity could be facilitated by activity of other cells physiologically independent of proteases (for example, the abstract is written as if this were not a possibility).
 - d. Claims on collagen IV crosslinking: The authors do conduct a nice collagenase assay followed by western blotting to show that there are dimers and monomers indicating some degree of covalent crosslinking type IV collagen to other collagen IV molecules. However, their claim that "Using this approach, the type IV collagen network in basement membrane explants is shown to be dominated by covalent cross-links (Fig 1E), thereby confirming that tissue-invasive cells confront a mechanically stable barrier" is flawed for several reasons. First, the high presence of monomers suggests a large fraction of collagen IV molecules are not in fact covalently crosslinked at their NC1 domain (unless I am missing something). Second, this kind of analysis does not reveal any insight into weak bonds along the side of col IV or apparently covalent crosslinks between 7S domains (expected to form

tetramers), which is required for network connectivity. For example, at an extreme end, if you have each col IV molecule covalently crosslinked to 1 other col IV molecule, you would not have a membrane but a solution of dimers. This analysis only shows the presence of covalent crosslinks, and not that they dominate the network, or that the network is a mechanically stable barrier (whatever this means).

e. Claims on BM mechanics: related to the last point, the literature on BM mechanics is mixed, and the authors do not conduct any mechanical testing in this study. Therefore, the authors should be careful about their commenting about the mechanics of the BM, and clarify which of their comments are supported, and use terms like “likely” when making claims like “the network is a mechanically stable barrier”.

2. Many of the results need to be supported by additional quantitative analysis with statistical testing. For example, figure 1 has no quantitative analysis! Statistical test details should be specified for each figure (number of cells; number of biological or technical replicates should be specified; also statistical test – authors say “Anova”, but what post-hoc test is used?). Some specific analyses that should be included are:

- Transmigration percentages must be quantified for data and experiments in Figure 1, 2, 3, 4, 5, and 6, since that is the critical biological behavior being studied in this manuscript. Further, how the authors quantify transmigration should be detailed in the methods section: I would think that a cell cannot be considered to have transmigrated until the cell has completely cleared the basement membrane layer. Related to this point, there is not really a clear set of time-lapse images showing macrophages transmigration completely through the BM layer. This should be included in Figure 1.
- Quantify pore dynamics of macrophage invasion through BM in the different conditions: The authors should quantify the pore size as macrophages transmigrates both with and without protease inhibitors or genetic knockdowns. This would provide some important mechanistic insight for macrophage invasion (beyond the actomyosin inhibition study), and reveal how large pores must be for macrophages to transmigrate, how much macrophages can dilate pores mechanically, and confirm that the pore dilation is elastic (and not plastic) in the case of protease inhibition, as the authors contend. Control analyses should be conducted on pores away from where macrophages invade through, and a sufficient number of cells should be analyzed for this (at least 10 but ideally many more so that there is confidence in the numbers.). This would be a very nice addition to the paper.
- Pore sizes should be quantified broadly at 0 hrs (to confirm what the pores sizes are prior to any degradation), and 48 hours with and without macrophages and cancer cells.
- The analysis of basement membrane architecture is really quite minimal here. Laminin and collagen IV images appear to be overexposed, and mechanical properties are unknown. Can the authors adjust the brightness and contrast to show any architectural features, and include a histogram of pixel intensity to show how much variation there is?

Minor comments:

1. There was a nice recent paper from Chavrier group nicely showing how MT1-MMP is involved in invasion independent of its proteolytic activity (Ferrari, et al., Nature Comm 2019). This should be referenced and noted in discussion.
2. Was cell seeding density controlled for in the macrophage experiments (e.g Fig. 5B +LPS mmp+/+ appears to have fewer cells compared to Fig 5B + LPS mmp-/-) and in the cancer cell experiments? Cancer cell images (Fig. 7C,E) appear to have lower cell density compared to macrophage experiments (Figs. 1I, 2B, 3A, 4A etc).
3. How do the authors reconcile current understanding that IL4-dependent polarization is linked to a remodeling phenotype and a phenotype that promotes cancer invasion and metastasis (an M2 or TAM phenotype) with the findings here that the IL4 cells demonstrate less remodeling, compared to LPS stimulated macrophages, suggesting the latter might restrict cancer invasion and metastasis? This should be discussed.
4. It is unclear where the macrophage membrane protrusions referred to in Fig. 1H are.
5. Why wasn't the transcriptional profiling in Fig. 3 not done under similar conditions as the rest of the experiments (6 days vs 1 day)?
6. It is difficult to see how there is an increase in cell surface-associated MT1-MMP upon LPS

treatment in Fig. 6b compared to control or IL4 treatment. Can authors provide some kind of quantification?

7. Provide description of alternative proteolytic systems treatment in Fig. 7B.

8. Typo on page 11: Fig 7AB should be Fig. 7A,B.

9. On page 23, the authors say, "...transmigrate both basement membrane and stromal tissue barriers..." This study was focused on basement membrane and strong evidence was not provided regarding transmigration across stromal tissue barriers. Therefore the reference to stromal tissue barriers should be removed.

10. It seems LPS treatment upregulated MRC1 in Supplementary Table 2, which is opposite what is described on page 16 of manuscript and Fig. 2A. Please clarify.

11. Why was the gene expression analysis in Fig. 9 compared to plastic? Plastic was not used in other experiments and it possibly has very different mechanical and biochemical properties compared to the decellularized mesentery.

12. A general comment is that this paper is not written as concisely as it could be in terms of the figures, with 9 figures. One suggestion for the authors to consider is to focus the manuscript figures on results specifically related to the points they make in the abstract, and move the other figures to the supplemental information. For example, Fig. 4, 5, and 9 are nice but don't seem to be essential for the main points of the manuscript. While it is certainly up to the authors how they want to present their manuscript, I would think their manuscript would make much more of an impact on the reader if it was more tightly written and organized around the most important findings.

Reviewer #3 (Remarks to the Author): with expertise in immune cell migration and invasion

The paper contains some beautiful work looking at the modes of macrophage and cancer cell migration into an ex-vivo basement membrane model. The study is well written and was a pleasure to read while WFH.

From the earlier figures looking at degradation it is unclear just how many of the unactivated macrophages are able to infiltrate the basement membrane (I am presuming they can use the pores), compared to the LPS activated macrophages that have higher levels/more MT1-MMP at the surface. I am assuming you can extract this data from these experiments as I am cognizant that the ability to perform more experiments is somewhat halted for many at the moment.

The final statement needs some further discussion: 'Importantly, this dual ability allows macrophages to specifically modulate transcriptional responses when infiltrating tissues and proteolytically remodeling the ECM in association with tissue-destructive events versus using mechanical forces in order to purposefully leave tissues unscathed during reparative states.'

The more pro-inflammatory gene signature macrophages versus the more M2 signature seen in the protease dependent vs protease independent migratory macrophages, respectively, are in LPS activated macrophages and typically LPS upregulates MT1-MMP. It is an interesting observation. How does this fit together with what might be happening in vivo? I am trying to wrap my head around when the latter scenario might occur i.e. activated cells entering the basement membrane but with little to no MT1-MMP. Perhaps some further discussion around this?

Am I right in my reading of the manuscript that you also have array data for unactivated macrophages +/- MT1-MMP? Although they are mouse, how does this compare to those macrophage profiles seen in Figure 9?

Figure 9 it is unclear from the figure legend the exact timing of activation, i.e. LPS added to cells then cells immediately seeded or where they activated for an amount of time before then? Can you please expand in the legend as the experimental detail is not clear?

One very minor comment is that final results section would benefit from a more detailed explanation of

the experiments performed to look at the transcriptome of cells traversing the basement membrane.

Reviewer #1 (Remarks to the Author): with expertise in cancer cell migration and invasion

Macrophages are known to infiltrate interstitial tissues during host defense mechanisms and inflammation. In the present work, using a decellularized mesentery extracellular matrix (ECM) tissue system that is mastered by this group, the authors addressed the mechanisms of basement membrane (BM) remodeling by mouse and human primary macrophages. Similar to previous work from this lab using several cellular systems including cancer-derived cell lines, the authors report that macrophages are able to remodel the BM in a MT1-MMP/MMP14-dependent manner. In addition, they found that LPS stimulation induces the expression of several proteinases in polarized macrophages, which correlates with a robust increase in their BM-remodeling capacity. However, despite mounting a strong proteinase response, polarized macrophages migrate across the BM in a proteinase- and MT1-MMP-independent and actomyosin-dependent manner, in sharp contrast with cancer cells that rely strictly on the MT1-MMP-dependent program for BM transmigration. Finally, MMP-dependent remodeling of the BM by LPS-stimulated macrophages is shown to induce a full transcriptional program possibly linking ECM remodeling and MT1-MMP function to host defense and inflammatory responses.

Overall, results are very neat and clear-cut, however some data regarding the dependency of the BM remodeling program on MT1-MMP are largely confirmatory of previous findings and long-established knowledge by this group based on different cellular models. In addition, some issues should be addressed in order to strengthen the manuscript, which are listed below (in the order of the figures).

1- Based on data such as in Figure 11, it seems that macrophages opportunistically make use of and widen preexisting pores in ex vivo mesentery BM, raising the issue of whether macrophages can actually pierce an intact BM and whether the observations are relevant to the in vivo situation.

This is an interesting question. However, the literature has consistently documented the fact that *all* basement membranes – from lung and skin to colon – are distinguished by pre-existing pores, allowing for both epithelial-mesenchymal crosstalk as well as tissue trafficking (e.g., as cited in the text, Barriero et al, 2016; Bluemink et al, 1976; Howat et al, 2001; Oakford et al, 2011; Takahashi-Iwanaga et al, 1999; Takeuchi et al, 2004). Further, several of these works document immune cell trafficking through these structures (ibid), including more recent studies that include elegant imaging of macrophage-basement membrane trafficking (e.g., *Cell Mol Gastro Hepatology* 12:1617, 2021). Complementing these findings, Paul Martin and colleagues performed live-imaging of macrophage-basement membrane transmigration in zebrafish during tumor initiation and likewise reported both proteolytic and non-proteolytic trafficking through pore-like structures (*Cell Reports* 27:2837, 2019). Given that i) intact peritoneal tissues are similarly decorated with basement membrane pores (see Figure 7) and ii) both macrophages and cancer cells cross the peritoneal basement membrane (e.g., *Cell* 165:668, 2016; *Cancer Metastasis Rev* 31:397, 2012; *Front Oncology* 7:24, 2017) we contend that our findings are relevant to the

***in vivo* situation. We also note that we have reported previously that cancer cell MMPs are likewise required to cross human dermal basement membranes (*Genes & Dev* 20:2673, 2006), the live chick chorioallantoic membrane basement membrane (*PNAS* 106:20318, 2009) and the mouse mammary gland basement membrane in a syngeneic breast cancer model (*Dev Cell* 47:145, 2018). Finally, as macrophage transmigration is increased in our system when it occurs concurrently with basement membrane degradation, we posit that creating basement membrane-‘free’ zones allows trafficking rates to increase beyond that provided by pores alone. Indeed, consistent with this proposition, basement membrane transmigration occurs more rapidly when macrophages are able to mobilize MMP activity (see Supplementary Figure 4A).**

2- Perforation of the basal BM in mesentery tissue seems to be a rare event based on the images provided in Figure 2B. If relevant, these events should be carefully quantified.

We have quantified the number of macrophages transmigrating and/or perforating the basal BM surface and have inserted these data in the revised text. At the 6 d time point, approximately 30% of the invading cell population has crossed the surface of the underlying basement membrane (see Supplementary Figure 4). To clarify, we emphasized this observation as recent studies have demonstrated that the BM is asymmetrically assembled with laminin facing the epithelial/mesothelial surface and type IV collagen localizing to the stromal side (*FEBS J* 282:4466, 2015). As such, our intention was only to demonstrate that macrophages applied to the surface of the peritoneum traffic across laminin before encountering the type IV collagen-rich layer and entering the stroma. Once within the stroma, the subset of macrophages that continue migrating would then first interface with the type IV collagen layer prior to traversing the basal BM surface, thereby demonstrating that macrophages can cross BMs in either direction.

3- Along similar lines, the authors state that macrophages were seen to encircle some apical BM perforations with no sign of protrusion or to send small protrusions through the cavity. Based on Video 5 and Video 6, again these seem to be rare events and they should be quantified. In addition, matrix distortion events and evidence for local force production are not so clear from the movies and these should be highlighted and quantified.

In attempting to quantify the number of macrophages encircling vs protruding through BM perforations, we encountered difficulty as the assays are standardly performed over longer time periods where macrophage bleaching becomes

problematic. As such, it is difficult to determine whether an ‘encircling’ macrophage might later access the pore or conversely, whether a protruding macrophage might - at some point – withdraw the cell body from the perforation and retreat back to the BM surface. Given these limitations, we have deleted Videos #5 and #6. However, with regard to matrix distortion, we have now quantified changes in basement membrane pore size during trafficking as outlined in the revised text both in the presence and absence of actomyosin-generated forces (see Figures 7 and 8).

4- Based on the immunoblotting analysis provided in Figure 5A, MT1-MMP levels are down-modulated in IL-4 treated macrophages. Yet, BM degradation of IL-4 treated vs control macrophages is similar (Figure 2E), raising the issue of the implication of MT1-MMP in BM remodeling by IL-4 polarized cells.

We have re-examined this issue as analyses of MT1-MMP expression in IL-4-treated mouse macrophages were performed with cells cultured atop a plastic tissue culture substratum. When we repeated the experiments atop the BM itself, the IL-4-treated macrophages *do* express MT1-MMP, though less than that observed for LPS-treated cells. Further, surface-associated MT1-MMP is only increased above control levels with LPS (see revised Figures 4A and 5A-C).

5- While both macrophages and breast tumor cells require MT1-MMP for BM remodeling, data in figure 7 show that these cell types differ in their requirement for MT1-MMP for transmigration through the BM; i.e. migration of macrophages through the BM is MT1-MMP independent in contrast to the strict requirement of tumor MDA-MB-231 and HT1080 cells for MT1-MMP. In addition, the migration of macrophages through the BM is nicely shown to require actomyosin contractility. These are rather preliminary observations that should be strengthened by analyzing the effect of blebbistatin/Y27632 in more detail for example looking at the effect of these compounds on earlier described phenotypes such as perforation encirclement and protrusion formation using live cell imaging. It would also be interesting to test the effect of actomyosin inhibitors on the transmigration of cancer cells through BM in the mesentery tissue model system in order to assess the generality or specificity of the actomyosin dependency.

We agree and have expanded our analyses by monitoring the effects of the inhibitors on protrusion formation as well as basement membrane remodeling (see revised Figure 8). With regard to cancer cells, we find that in the absence of protease inhibitors, neither Y27632 nor blebbistatin prevent basement membrane degradation or invasion (see Supplementary Figure 4). These findings are likely consistent with the fact that in the face of wholesale basement membrane degradation by cancer cells, major distortions of cell shape during transmigration are not required.

Reviewer #2 (Remarks to the Author): with expertise in cancer cell migration and invasion

Bahr et al. present a study investigating protease dependent and independent interactions between human/mouse macrophages or HT-1080/MDA-MB231 cancer cells with native matrix explants. They conclude that macrophages and cancer cells rely on MT1-MMP for basement membrane remodeling and transmigration. However, they find that while macrophages are able to transmigrate across the basement membrane in the presence of protease inhibitors, cancer cells wholly depend on protease activity for transmigration.

This manuscript presents some compelling and important ideas and results utilizing a powerful model system, however, major revisions are required for this manuscript to become suitable for publication in Nature Communications. The model system of basement membrane invasion is quite powerful, and the finding that macrophages can traffic through basement membranes independent of proteases is an important new finding. The potential finding of protease-dependent invasion required for invasion of cancer cells through rat mesentery BM is important and adds to the current literature in this area. There are also some interesting experiments looking at macrophage activation and phenotype and the impact on remodeling. However, there are some major issues with this manuscript that need to be addressed. Many of the claims presented in this manuscript are vastly overstated and need to be moderated. Much of the analysis is qualitative and conclusions based on representative images; additional rigorous quantitative analysis is required for many of the results. Finally, some additional mechanistic insight into pore dynamics during macrophage invasion would make this stronger.

However, there are some major issues with this manuscript that need to be addressed. Many of the claims presented in this manuscript are vastly overstated and need to be moderated. Much of the analysis is qualitative and conclusions based on representative images; additional rigorous quantitative analysis is required for many of the results. Finally, some additional mechanistic insight into pore dynamics during macrophage invasion would make this stronger.

Each of these issues have been addressed in the revised manuscript. Requested quantitative analyses are included and additional mechanistic insights into pore dynamics provided (see below).

Major comments:

1. Some of the claims made in this manuscript are far too broad, and either need to be moderated, or supplemented with additional experiments or analyses that better justify their broad claims.

a. Extrapolation of claims beyond mesentery basement membranes: In a number of places, the authors generalize their results to make conclusions about basement membrane remodeling and invasion very broadly. However, they have done studies in only one basement membrane system – rat mesentery basement membrane. It is well

known that the composition (e.g. type of laminin), architecture (e.g. thickness), and mechanical properties of basement membrane vary widely in humans. Rat mesentery basement membrane is not representative of all human BMs, and I don't believe that mesentery BM is even traversed in most cancers. Further, the BM is prepared in ways that could potentially impact the structure of the rat mesentery BM. Thus, the claims like "By contrast, cancer cell invasive activity is completely reliant on metalloproteinase activity and neither mechanical force nor changes in nuclear rigidity rescue transmigration" or "Hence, in contrast to macrophages, human cancer cells are entirely reliant on MMP activity to transmigrate native tissue barriers", and similar statements, need to be qualified, and the fact that rat mesentery BM is studied here should be specified in the abstract. If the authors wanted to make broader claims, they would have to show these results in other physiological models of BM (for example BM of vasculature might be possible).

The Reviewer has raised a number of interesting points. Firstly, as the Reviewer correctly points out, all BMs are 'different' with regard to "composition (eg, type of laminin), architecture, and mechanical properties..", but the key issue that we wish to stress is that a type IV collagen-containing, covalently cross-linked BM is widely considered to be the final barrier that all transmigrating cells must negotiate - regardless of the tissue site. We have previously validated all of our previous studies on cancer cell-BM trafficking by comparing the peritoneal model with human dermal explants as well as the live chick chorioallantoic membrane model and more recently, in the mouse mammary gland *in vivo* (*Genes & Dev* 20:2763, 2006; *PNAS* 106:20318, 2009; *Dev Cell* 47:145, 2018). While we have gone to some effort through the years to validate our findings in multiple systems, we note that the highly cited works by the Sherwood group on basement membrane invasion (e.g., *PNAS* 115:11537, 2018; *Dev Cell* 48:313, 2019; *ibid* 57:732, 2022) or the recent works published in *Nat Commun* by the Vignjevic or Chaudhuri groups (*ibid* 8:924, 2017; 9:4144, 2018) have based their conclusions on the use of either a single BM model or synthetic BM mimics, respectively.

Secondly, we wish to clarify the fact that the mesenteric basement membrane (BM) is indeed traversed in many carcinomatous states, ranging from ovarian and pancreatic carcinomas to colon cancer (*Cancer Metastasis Rev* 31:397, 2012; *Front Oncol* 7:24, 2017). Similarly, macrophages also cross this barrier in pathological states (*Cell* 165:668, 2016). With regard to the 'potential' ability of cancer cells to transmigrate BMs independently of proteolytic activity, these conclusions are based on the use of 3 systems; *C elegans* anchor cell invasion (*PNAS* 115:11537, 2018; *Dev Cell* 48:313, 2019), a peritoneal basement membrane model similar to the one used here (*Nat Commun* 8:924, 2017), and *in vitro* constructs using a 'reconstituted' extract of basement membrane proteins (*Nat Commun* 9:4144, 2018; *Matrix Biol* 85-86:94, 2020). Only the work with *C elegans* and the peritoneal model rely on the use of 'authentic' basement membranes. While these works are routinely cited in recent reviews as

providing compelling evidence for the ability of "... physical forces generated by cancer cells facilitating protease-independent BM invasion." (eg, see Chang & Chaudhuri, *J Cell Biol* 218:2456, 2019), none of these cited works characterized either the covalent structure or mechanical properties of authentic BMs. Even more problematic, commonly used BM 'extract'-based models employ Matrigel, a composite long known to poorly recapitulate the structural or mechanical properties of the critical network of cross-linked type IV collagen molecules found in authentic BMs (*Trends Cell Biol* 18:560, 2008). We also stress that the BM mechanical integrity is dependent on the number of covalent sulfilimine crosslinks, a characteristic largely ignored in each of the works cited above. Extensive studies published by Hudson and colleagues on BM crosslinking (eg, *Nat Chem Biol* 8:784, 2012; *Cell* 157:1380, 2014) have clearly established the major role played by these structures in maintaining the mechanical stability of BMs *in vivo*. Nevertheless, we have tempered our conclusions as the Reviewer suggests, while more clearly stressing the application of similar caveats to the work of others in the field.

b. Claim about protease dependence of cancer cell invasion: In this assay, cancer cells could simply continue migrating along the plane of the BM, where migration is unimpeded (in contrast to native tissue environment around a BM), as migrating cells often follow paths of least resistance. The cells don't necessarily have to invade through the BM if they are not driven/forced to (for example by a strong chemokine gradient). Thus, the lack of invasion of cancer cells through the BM, doesn't mean that it is not possible – it may be that the cell is just not activated sufficiently to do so. This is why one has to be careful making such strong definitive statements that cancer cells cannot invade through BM at all, independent of proteases.

This is an interesting point, but investigators in the field do not routinely classify 2D migration across an ECM surface as 'invasion' as carcinomas must access stromal tissues in order to intravasate vascular or lymphatic beds in order to metastasize. Nevertheless, the Reviewer is correct in pointing out that the term 'invasion' should be used more carefully. As such, we have revised the text to more clearly indicate that we are describing the mechanisms by which cells penetrate the BM to access the stromal compartment. Finally, we have inserted new data demonstrating that cancer cells – even when growth factor-triggered are unable to cross the BM (Figure 9D and Supplementary Figure 5). This work also agrees with our earlier studies (cited above in *Genes and Dev* 2006 and *PNAS* 2009) demonstrating that protease-inhibited cancer cells are unable to cross peritoneal or dermal BMs *in vitro* or the live chick embryo BM *in vivo*. Certainly, the *possibility* exists that cancer cells could penetrate BMs independently of proteolytic activity, but we are unaware of where this activity has been documented using authentic BM constructs. Indeed, a recent report in *Nature* 582:253, 2020 likewise describes a critical - and required - role for MT1-MMP in BM remodeling during embryogenesis.

c. Even if the authors are able to more strongly support their claim in b, these data are not inconsistent with the findings of Glentis, et al. that CAFs open up pores for cancer cells to invade through. That is still a distinct possibility and it should be acknowledged that cancer cell invasive activity could be facilitated by activity of other cells physiologically independent of proteases (for example, the abstract is written as if this were not a possibility).

We agree that a possibility exists. However, we can only reference the work by Glentis et al with caution as the fibroblasts were actually seeded artificially on the basement membrane itself rather than embedded in the stromal tissues *beneath* and between the two basement membranes. We have modified our text, but the ability of stromal fibroblasts to widen basement membrane pores to allow cancer cell transmigration remains to be demonstrated.

d. Claims on collagen IV crosslinking: The authors do conduct a nice collagenase assay followed by western blotting to show that there are dimers and monomers indicating some degree of covalent crosslinking type IV collagen to other collagen IV molecules. However, their claim that “Using this approach, the type IV collagen network in basement membrane explants is shown to be dominated by covalent cross-links (Fig 1E), thereby confirming that tissue-invasive cells confront a mechanically stable barrier” is flawed for several reasons. First, the high presence of monomers suggests a large fraction of collagen IV molecules are not in fact covalently crosslinked at their NC1 domain (unless I am missing something). Second, this kind of analysis does not reveal any insight into weak bonds along the side of col IV or apparently covalent crosslinks between 7S domains (expected to form tetramers), which is required for network connectivity. For example, at an extreme end, if you have each col IV molecule covalently crosslinked to 1 other col IV molecule, you would not have a membrane but a solution of dimers. This analysis only shows the presence of covalent crosslinks, and not that they dominate the network, or that the network is a mechanically stable barrier (whatever this means).

First, type IV collagen knockout mice have been shown to deposit a basement membrane that lacks mechanical stability and gives way to an embryonic lethal phenotype (*Development* 131:1619, 2004). Second, unlike lysyl oxidase-derived 7S crosslinks, only NC1 sulfilimine crosslinks are known to play a required role in maintaining the mechanical stability of basement membranes *in vivo* (*Cell* 157:1389, 2014; *Am J Physiol Renal Physiol* 313:F596, 2017). Indeed, *LOXL2* knockout mice (the LOX isoform associated with 7S cross-links; *JBC* 291:2599, 2016) are viable and fertile (*EMBO J* 34:10190, 2015). In mammals, the highest density of NC1 cross-links are found in the placental and glomerular basement membranes with the lowest densities confined to the lens capsule (presumably due its need to accommodate shape changes in the lens) (*Conn Tissue Res* 55:8, 2014). While these findings underline the range of NC1 crosslinks found in

mammalian tissues, we are unaware of any literature suggesting that the number of LOX-derived, 7S cross-links vary between tissues. In any case, the levels of NC1 cross-links detected in the peritoneum are similar to the high levels found in the placenta (*Conn Tissue Res* 55:8, 2014). As the structure of type IV collagen cross-link formation is complex, we should better stress the fact that the isolated 'dimers' are not intramolecular in origin, but rather intermolecular dimers containing either 1 or 2 sulfilimine cross-links, respectively, between the NC1 domains of two type IV heterotrimers (i.e., hexamers) (*Methods Cell Biol* 143:171,2018). Given that the number of sulfilimine cross-links found in the peritoneum are similar to those found in tissues harboring the maximal number of cross-links and that the basement membrane cannot be penetrated by cancer cells (even in the presence of a strong chemotactic signal), we hold that the barrier can at least be qualified as mechanically stable (and not plastic) – at least from a biologic perspective.

e. Claims on BM mechanics: related to the last point, the literature on BM mechanics is mixed, and the authors do not conduct any mechanical testing in this study. Therefore, the authors should be careful about their commenting about the mechanics of the BM, and clarify which of their comments are supported, and use terms like "likely" when making claims like "the network is a mechanically stable barrier".

Agreed.

2. Many of the results need to be supported by additional quantitative analysis with statistical testing. For example, figure 1 has no quantitative analysis! Statistical test details should be specified for each figure (number of cells; number of biological or technical replicates should be specified; also statistical test – authors say "Anova", but what post-hoc test is used?). Some specific analyses that should be included are:

-Transmigration percentages must be quantified for data and experiments in Figure 1, 2, 3, 4, 5, and 6, since that is the critical biological behavior being studied in this manuscript. Further, how the authors quantify transmigration should be detailed in the methods section: I would think that a cell cannot be considered to have transmigrated until the cell has completely cleared the basement membrane layer. Related to this point, there is not really a clear set of time-lapse images showing macrophages transmigration completely through the BM layer. This should be included in Figure 1.

We have included all requested information and also completely reorganized the text. The manuscript was meant to address 3 questions; i) can macrophages proteolytically degrade the BM and by what mechanism, ii) are macrophages able to traverse BMs, and if so, by protease- dependent and/or independent processes and iii) how does the ability of macrophages to degrade and traverse BMs compare/contrast with that of cancer cells? With regard to BM degradation

and transmigration, all of the statistical analyses for the area of BM degraded and the changes in perforation size are now included in the text. We erred in organizing the original manuscript by prematurely including data on 'invasion/transmigration' in Fig. 2B and 3A when quantitative data pertinent to these variables were not included until Figures 7 and 8. In the revised text, the first set of figures focus solely on BM degradation – along with all quantitative data and statistical analyses, while the second set include all quantitative data and analyses for transmigration in either macrophage or cancer cell populations.

With specific regard to Fig.1, the data are presented only to highlight the organization of the construct, the status of type IV collagen crosslinks, and the ability of the macrophages to create perforations. More specific details regarding the % BM area degraded and perforation size and the effects of LPS vs IL-4 stimulation are shown in Fig. 2. As described above, all findings relevant to invasive activity have been moved to the latter half of the manuscript along with all requested statistical analyses.

Finally, we have included images of tissue explants that clearly show that macrophages have transmigrated completely through the BM surface and rest within the underlying interstitial compartment (see Figure 6B). While we have labeled live macrophages to more clearly highlight trafficking across the BM in real-time, the process is sufficiently slow (on the order of 6 h or more) that bleaching precludes imaging of the entire process, though we do show that macrophages completely clear the upper surface as they embed themselves in the stromal compartment. Only cells whose nuclei lie beneath the apical face of the basement membrane are classified as having transmigrated. Additional detail regarding 'transmigration' have been included in the revised Methods as well.

-Quantify pore dynamics of macrophage invasion through BM in the different conditions: The authors should quantify the pore size as macrophages transmigrates both with and without protease inhibitors or genetic knockdowns. This would provide some important mechanistic insight for macrophage invasion (beyond the actomyosin inhibition study), and reveal how large pores must be for macrophages to transmigrate, how much macrophages can dilate pores mechanically, and confirm that the pore dilation is elastic (and not plastic) in the case of protease inhibition, as the authors contend. Control analyses should be conducted on pores away from where macrophages invade through, and a sufficient number of cells should be analyzed for this (at least 10 but ideally many more so that there is confidence in the numbers.). This would be a very nice addition to the paper.

As requested, we have monitored pore size changes during macrophage transmigration, including the absence of pore size in control basement membranes as well as in pores located at sites distant from invading cells (see Figures 7 and 8 as well as Supplementary Figure 3. These measurements have allowed us to demonstrate that pore dilation is primarily elastic in nature.

However, with regard to protease-dependent invasion, the macrophages irreversibly degrade large swathes of the basement membrane whose size far exceeds that of individual cells prior to invasion, thereby precluding efforts to monitor changes in pore size coincident with invasive activity (i.e., cells do not appear to proteolytically ‘burn’ small holes in the basement membrane to access the underlying stroma).

-Pore sizes should be quantified broadly at 0 hrs (to confirm what the pores sizes are prior to any degradation), and 48 hours with and without macrophages and cancer cells.
-The analysis of basement membrane architecture is really quite minimal here. Laminin and collagen IV images appear to be overexposed, and mechanical properties are unknown. Can the authors adjust the brightness and contrast to show any architectural features, and include a histogram of pixel intensity to show how much variation there is?

We apologize for not making this more clear, but pore sizes – both prior to migration and in the native, non-decellularized BM – were shown in the original text. Basement membrane pores do not change in size under control conditions (see above). We have also expanded these analyses and included the requested measurements in the revised text (see Figures 7,8 and Supplementary Fig 3).

With regard to image intensity, we have purposefully adjusted the images carefully to avoid overexposure. We have also included normalized fluorescence line intensity analyses in the revised text to characterize changes in BM staining intensity and the characteristics of proteolyzed and non-proteolyzed pores (see Figures 2 and 7 as well as Supplementary Fig 3).

Minor comments:

1. There was a nice recent paper from Chavrier group nicely showing how MT1-MMP is involved in invasion independent of its proteolytic activity (Ferrari, et al., Nature Comm 2019). This should be referenced and noted in discussion.

We agree and have included the work in the revised text. It should be noted, however, that we do not see MT1-MMP-dependent invasion independent of its proteolytic activity (similar to that described by Chavrier) as MT1-MMP-null macrophages successfully transmigrate the basement membrane.

2. Was cell seeding density controlled for in the macrophage experiments (e.g Fig. 5B +LPS mmp+/+ appears to have fewer cells compared to Fig 5B + LPS mmp-/-) and in the cancer cell experiments? Cancer cell images (Fig. 7C,E) appear to have lower cell density compared to macrophage experiments (Figs. 1I, 2B, 3A, 4A etc).

The seeding density of macrophage as well as cancer cells were held constant at 1×10^5 cells/BM. As the distribution of cells atop the BM is not homogenous (i.e.,

the cells are not seeded at high density to create confluent monolayers) and the images randomly selected, the 'apparent' cell density is different in specific images.

3. How do the authors reconcile current understanding that IL4-dependent polarization is linked to a remodeling phenotype and a phenotype that promotes cancer invasion and metastasis (an M2 or TAM phenotype) with the findings here that the IL4 cells demonstrate less remodeling, compared to LPS stimulated macrophages, suggesting the latter might restrict cancer invasion and metastasis? This should be discussed.

This is an interesting observation, but while M2/TAM-type macrophages can clearly play an important role in tumor progression, we are unaware of data demonstrating that their ability to promote invasion/metastasis are linked to their matrix-remodeling activity as opposed to the secretion of tumor growth factors, cytokines, miRNA-rich exosomes, etc.

4. It is unclear where the macrophage membrane protrusions referred to in Fig. 1H are.

The image shown is the lower face of the upper basement membrane with penetrating macrophage protrusions (green) shown. We have clarified this point in the revised text.

5. Why wasn't the transcriptional profiling in Fig. 3 not done under similar conditions as the rest of the experiments (6 days vs 1 day)?

As perforations are readily seen within 24 h of culture, we limited our initial analyses to the earlier time point.

6. It is difficult to see how there is an increase in cell surface-associated MT1-MMP upon LPS treatment in Fig. 6b compared to control or IL4 treatment. Can authors provide some kind of quantification?

Image has been quantified as requested in Fig 5C.

7. Provide description of alternative proteolytic systems treatment in Fig. 7B.

This cocktail includes general inhibitors directed against metallo-, serine-cysteine- and aspartyl proteinases (*JCB* 201:1069, 2013) and the description has been added to the text.

8. Typo on page 11: Fig 7AB should be Fig. 7A,B.

Corrected.

9. On page 23, the authors say, "...transmigrate both basement membrane and stromal tissue barriers..." This study was focused on basement membrane and strong evidence was not provided regarding transmigration across stromal tissue barriers. Therefore the reference to stromal tissue barriers should be removed.

As shown in Fig. 1B of the original text, the reflected BM is subtended by a type I collagen and elastin-rich stromal matrix. As macrophages (as well as carcinoma cells) reach and perforate the lower BM (quantified in the revised text), this can only be accomplished if the cells moved through the stromal compartment. We now include an image of 'interstitial' macrophages in Fig 6B.

10. It seems LPS treatment upregulated MRC1 in Supplementary Table 2, which is opposite what is described on page 16 of manuscript and Fig. 2A. Please clarify.

The columns in Supplementary Table 2 were reversed. We attempted to alert the editors during the review to advise the referees, but the message was apparently lost.

11. Why was the gene expression analysis in Fig. 9 compared to plastic? Plastic was not used in other experiments and it possibly has very different mechanical and biochemical properties compared to the decellularized mesentery.

We are aware of the fact that the mechanical/biochemical properties of plastic surfaces are distinct from that of the ECM constructs. However, virtually all transcript profiling in the literature – including descriptions of M1 vs M2 genotypes, have been performed in macrophages atop tissue culture-grade plastic. Hence, we used this surface as the 'control' and despite the differences, noted that more than 400 transcripts were similarly expressed across these very different substrata.

12. A general comment is that this paper is not written as concisely as it could be in terms of the figures, with 9 figures. One suggestion for the authors to consider is to focus the manuscript figures on results specifically related to the points they make in the abstract, and move the other figures to the supplemental information. For example, Fig. 4, 5, and 9 are nice but don't seem to be essential for the main points of the manuscript. While it is certainly up to the authors how they want to present their manuscript, I would

think their manuscript would make much more of an impact on the reader if it was more tightly written and organized around the most important findings.

We agree, and the point is well-taken. We have reorganized the entire text and done our best to sharpen the focus as suggested.

Reviewer #3 (Remarks to the Author): with expertise in immune cell migration and invasion

The paper contains some beautiful work looking at the modes of macrophage and cancer cell migration into an ex-vivo basement membrane model. The study is well written and was a pleasure to read while WFH.

From the earlier figures looking at degradation it is unclear just how many of the unactivated macrophages are able to infiltrate the basement membrane (I am presuming they can using the pores), compared to the LPS activated macrophages that have higher levels/more MT1-MMP at the surface. I am assuming you can extract this data from these experiments as I am cognizant that the ability to perform more experiments is somewhat halted for many at the moment.

Unactivated macrophages, rather than assuming a completely ‘resting’ state, are able to display considerable basement membrane degradative activity - approximately half of that recorded for LPS-stimulated cells(Fig 2C). Interestingly, they likewise traverse basement membranes to a degree similar to that found with LPS-stimulated macrophages – at least after 6 d of culture (i.e., $60.6 \pm 3.1\%$ vs $69.2 \pm 11.4\%$; mean \pm SEM, n=3). As up to 15% of the basement membrane is degraded by the untreated macrophages under these conditions, it is difficult for us to assign the degree to which macrophages cross newly generated perforations versus pre-existing portals. Nevertheless, we do find that in the absence of MMP-dependent proteolysis, the rate of macrophage infiltration is considerably slowed at 2-4 d of culture, despite the fact that the final degree of invasion is comparable by 6 d (see Supplementary Figure 4A).

The final statement needs some further discussion: ‘Importantly, this dual ability allows macrophages to specifically modulate transcriptional responses when infiltrating tissues and proteolytically remodeling the ECM in association with tissue-destructive events versus using mechanical forces in order to purposefully leave tissues unscathed during reparative states.’

The more pro-inflammatory gene signature macrophages versus the more M2 signature seen in the protease dependent vs protease independent migratory macrophages, respectively, are in LPS activated macrophages and typically LPS upregulates MT1-MMP. It is an interesting observation. How does this fit together with what might be happening in vivo? I am trying to wrap my head around when the latter scenario might

occur i.e. activated cells entering the basement membrane but with little to no MT1-MMP. Perhaps some further discussion around this?

This is an interesting issue. A recent example that might well apply to all tissues in general is the detection of intraepithelial macrophages that have crossed the normal mammary gland basement membrane (e.g., *Front Cell Dev Biol.* 7:250, 2019) or even the presence of macrophage protrusions crossing the epithelial basement membrane in normal colon (*Cell* 183:411, 2020) or the vascular basement membrane in skin (*eLIFE* 5:e15251, 2016). In each of these cases, given the normal structure of these basement membranes, we posit that macrophages traverse/infiltrate these barrier by accessing resident pores without engaging proteolytic activity while maintaining a quiescent (i.e., non-inflammatory) phenotype. By contrast, recent studies in *Drosophila* have shown that Mmp2-dependent basement membrane damage induces a pro-inflammatory response (*Nat Commun* 11:3631, 2020). We have included a discussion of this issue in the revised text.

Am I right in my reading of the manuscript that you also have array data for unactivated macrophages +/- MT1-MMP? Although they are mouse, how does this compare to those macrophage profiles seen in Figure 9?

We have only found partial overlap with these transcript profiles, but underscore the fact that the mouse studies were done at 24 h while the human macrophages were harvested at a 48 h time point. Distinct responses have previously been noted when comparing LPS-stimulated mouse vs human macrophages (*PNAS* 109:E944, 2012; *Front Cell Dev Biol* 8:1, 2020), but we also hesitate to directly compare macrophages generated from mouse bone marrow cells cultured with M-CSF to those generated from human peripheral blood and cultured in serum alone.

Figure 9 it is unclear from the figure legend the exact timing of activation, i.e. LPS added to cells then cells immediately seeded or where they activated for an amount of time before then? Can you please expand in the legend as the experimental detail is not clear?

We have included the experimental details in the legend where LPS is added immediately after seeding atop the explant. As an aside, when we attempted to stimulate the cells prior to placing the macrophages atop the explants, the LPS-triggered cells adhered to the plastic substratum so tightly that they could not be detached without major losses in viability.

One very minor comment is that final results section would benefit from a more detailed explanation of the experiments performed to look at the transcriptome of cells traversing the basement membrane.

We have expanded this portion of the text to make the details of the experimental design more clear.

REVIEWERS' COMMENTS

Reviewer #1 (Remarks to the Author):

The revised manuscript addressed all points to my satisfaction. I would therefore recommend publication of the manuscript reporting a well-conducted and exciting study.

Reviewer #2 (Remarks to the Author):

The authors have put in a significant effort into the revised manuscript, which is appreciated. This is an outstanding paper that should be published in Nature Communications. However, there are still a few claims that go beyond what the data show. The claims need to be revised accordingly.

1. The authors make the claim in the abstract that “cancer cell invasive activity is completely reliant on metalloproteinase activity”. This is not supported by the data. Fig. 9D shows a non-zero percentage of cells that transmigrate in the EGF/+BB-94 condition. Thus, some cells are inf act invading through the basement membrane in this case according to the data. Further, as pointed in the last review, cells may simply be migrating in the plane of the basement membrane and not sufficiently induced to invade through it (for example, the growth factor gradient might not be sufficiently strong to attract the cells to invade versus migrating in the plane of the basement membrane). While its possible the authors are correct in the assertion, and some of their critiques of previous studies are valid, the challenge is that it is very difficult to prove a negative. Such strong claims that are peripheral to the macrophage story and in my view detract from the story. Such claims should be moderated (“i.e. cancer cell activity is highly dependent” or similar) throughout the manuscript in the abstract and results, though the authors are free to speculate in the discussion.

2. It is not clear to me that the authors have supported the claim that these BMs are “dominated by covalent cross-links” (line 114). This assay doesn’t speak to the presence of weak bonds, if any, so its not clear whether one modality is more dominant than the other. Further, if the authors wish to claim that “levels similar to those found in other highly cross-linked basement membranes”, they should include the quantification of the Western blot of percent NC1 in monomer vs. dimer form (as was done in the referred to manuscript – ref. 32).

3. While the additional quantification of pore dynamics has been included, which is appreciated, it is not clear to me that the authors can conclude that the “pore dilation is primarily elastic in nature”. To show this what they would need is the pore-size before cell arrives, the pore size as the cell invades through, and the pore size after cell passes through. It looks like they just have two time points (during and after, but not before; Fig. 7D, and 7F), so while they can show some elastic recovery, it looks like there might also is some plastic deformation (Fig 7G). Unless the authors can get the full trajectory of the pore size dynamics, the claim should be moderated.

RESPONSES TO REVIEWERS

The authors have put in a significant effort into the revised manuscript, which is appreciated. This is an outstanding paper that should be published in Nature Communications. However, there are still a few claims that go beyond what the data show. The claims need to be revised accordingly.

1. The authors make the claim in the abstract that “cancer cell invasive activity is completely reliant on metalloproteinase activity”. This is not supported by the data. Fig. 9D shows a non-zero percentage of cells that transmigrate in the EGF/+BB-94 condition. Thus, some cells are in fact invading through the basement membrane in this case according to the data.

Our conclusion was based on the fact that in the presence of the requested growth factor gradient, cancer cell transmigration decreased from a range of 40-50% to less than 3% - a degree of inhibition that stands in direct contrast to claims of 'no inhibition' described in other reports (eg, Dev Cell 48:313,2019; Nat Commun 9:4144, 2018; Dev Cell 57:732, 2022). To claim that the very few cells that do invade are using a bonafide alternate system as opposed to 'simply' negotiating small tears in the basement membrane would likewise seem to be an over interpretation of the facts at hand. Nevertheless, the fact a non-zero percentage was recorded is true and we have modified the text accordingly.

Further, as pointed in the last review, cells may simply be migrating in the plane of the basement membrane and not sufficiently induced to invade through it (for example, the growth factor gradient might not be sufficiently strong to attract the cells to invade versus migrating in the plane of the basement membrane). While its possible the authors are correct in the assertion, and some of their critiques of previous studies are valid, the challenge is that it is very difficult to prove a negative. Such strong claims that are peripheral to the macrophage story and in my view detract from the story. Such claims should be moderated (“i.e. cancer cell activity is highly dependent” or similar) throughout the manuscript in the abstract and results, though the authors are free to speculate in the discussion.

We have used the most widely employed cancer cells, i.e., MDA-MB-231 and HT-1080 cells, wherein 'amoeboid/non-proteolytic' mechanisms have been popularized with regard to a range of non-physiologic barriers. We agree that proving a negative is difficult – if not impossible, but to alternatively contend that we have not tested every other experimental variable imaginable (e.g., different chemoattractants, different gradients, different cells, different basement membranes, etc.) would seem to hold us to a standard of negating every – and any flawed experiment or interpretation published in the literature. This is – at least to this author – akin to arguments forwarded in the lay press, i.e., 'prove me that I'm wrong'. We have no problem with softening the text to read "highly dependent" rather than 'completely dependent', but we do not wish to detract from the fact that the notion that cancer cells can simply switch from protease-dependent to an equally effective protease-independent mode of invasion is not supported by the data, at least with regard to a physiologic barrier.

2. It is not clear to me that the authors have supported the claim that these BMs are “dominated by covalent cross-links” (line 114). This assay doesn't speak to the presence of weak bonds, if any, so its not clear whether one modality is more dominant than the other. Further, if the authors wish to claim that “levels similar to those found in other highly cross-linked basement membranes”, they should include the quantification of the Western blot of percent NC1 in monomer vs. dimer form (as was done in the referred to manuscript – ref. 32).

As outlined in our last response, the literature is replete with works highlighting the importance of sulfilimine cross-links in maintaining the structural integrity of mechanically-loaded basement membranes in vivo. Nevertheless, the most commonly used basement membrane surrogate, Matrigel, is devoid of covalent cross-links, let alone those derived from sulfilimines. However, as we pointed out previously, basement membranes do exist in vivo that are characterized by extremely low levels of sulfilimine cross-links, such as that found in the optic lens. Our only intent was to show that the basement membranes we use are replete with cross-links. As such, we have included the quantitation (78% dimer/22% monomer), but I would stress that the onus for performing this experiment should be placed on those investigators that claim evidence of protease-independent modes of basement membrane invasion as the absence of these cross-links would alter interpretation.

3. While the additional quantification of pore dynamics has been included, which is appreciated, it is not clear to me that the authors can conclude that the "pore dilation is primarily elastic in nature". To show this what they would need is the pore-size before cell arrives, the pore size as the cell invades through, and the pore size after cell passes through. It looks like they just have two time points (during and after, but not before; Fig. 7D, and 7F), so while they can show some elastic recovery, it looks like there might also be some plastic deformation (Fig 7G). Unless the authors can get the full trajectory of the pore size dynamics, the claim should be moderated.

We apologize if we did not make this point more clear. As opposed to the two time point shown in Figs 7D and F, we then quantified changes in pore size in over 500 basement membrane perforations from time 0 through day 6 of macrophage transmigration in 3 independent experiments. Under these conditions, pore size, having expanded to a peak of approximately 9 μm^2 from 4 μm^2 then decreased back to 5 μm^2 – an 80% recovery of control basement membrane pore size. As we cannot rule out the possibility that a further decrease in pore size might have taken place with longer culture period, that the distribution of pore size might have been skewed by stable cell protrusions (e.g., Fig 8G) or even continued cell migration, we prefer to conclude that most of the observed change in pore size can be attributed to elastic recovery.

Bahr et al. have adequately addressed the critique of Reviewer #3. There are however additional issues that should be addressed:

1) the color images in this manuscript are striking, but it would seem that pseudo colored images should be used to insure that the images can be evaluated by those who are color-blind. The authors state in their rebuttal in regard to a question about overexposure of laminin and collagen IV images that "With regard to image intensity, we have purposefully adjusted the images carefully to avoid overexposure." Presenting such single color images in grey scale might help as to this reviewer's eye the images also appear to be overexposed.

In discussing this issue with the Editor, we have included a series of supplementary figures where the red and green channels have been separated. Further, with regard to overexposure, at least two issues are relevant here. First, please be advised that when viewing images generated through the entire explant, we are actually imaging through 2 separate basement membranes. Second, to address this concern raised earlier, we specifically included normalized fluorescence line intensity analyses to

characterize changes in normal BM staining intensity as well as the characteristics of proteolyzed and non-proteolyzed pores (see Figures 2 and 7 as well as Supplementary Fig 3).

In addition, including in the legends explanation of what the various colors indicate would help as a reader is faced with beautiful images that are not readily interpreted.

Agreed.

2) Panels E and F of Figure 3 should be modified. As presented in the Figure and described in the legend, these panels are said to be data on expression of proteases, yet both panel E and panel F include one serine protease receptor. In panel F, aspartic and serine proteases are designated as cysteine proteases. Similarly in the text in line 183, the authors use "cathepsin" to designate a proteinase family involved in ECM remodeling. This is not accurate as the name cathepsin is used for three proteinase families: aspartic, cysteine and serine.

Corrected- thank you for noting this oversight.

RESPONSES TO REVIEWERS

The authors have put in a significant effort into the revised manuscript, which is appreciated. This is an outstanding paper that should be published in Nature Communications. However, there are still a few claims that go beyond what the data show. The claims need to be revised accordingly.

1. The authors make the claim in the abstract that “cancer cell invasive activity is completely reliant on metalloproteinase activity”. This is not supported by the data. Fig. 9D shows a non-zero percentage of cells that transmigrate in the EGF/+BB-94 condition. Thus, some cells are inf act invading through the basement membrane in this case according to the data.

Our conclusion was based on the fact that in the presence of the requested growth factor gradient, cancer cell transmigration decreased from a range of 40-50% to less than 3% - a degree of inhibition that stands in direct contrast to claims of 'no inhibition' described in other reports (eg, Dev Cell 48:313,2019; Nat Commun 9:4144, 2018; Dev Cell 57:732, 2022). To claim that the very few cells that do invade are using a bonafide alternate system as opposed to 'simply' negotiating small tears in the basement membrane would likewise seem to be an over interpretation of the facts at hand. Nevertheless, the fact a non-zero percentage was recorded is true and we have modified the text accordingly.

Further, as pointed in the last review, cells may simply be migrating in the plane of the basement membrane and not sufficiently induced to invade through it (for example, the growth factor gradient might not be sufficiently strong to attract the cells to invade versus migrating in the plane of the basement membrane). While its possible the authors are correct in the assertion, and some of their critiques of previous studies are valid, the challenge is that it is very difficult to prove a negative. Such strong claims that are peripheral to the macrophage story and in my view detract from the story. Such claims should be moderated (“i.e. cancer cell activity is highly dependent” or similar) throughout the manuscript in the abstract and results, though the authors are free to speculate in the discussion.

We have used the most widely employed cancer cells, i.e., MDA-MB-231 and HT-1080 cells, wherein 'amoeboid/non-proteolytic' mechanisms have been popularized with regard to a range of non-physiologic barriers. We agree that proving a negative is difficult – if not impossible, but to alternatively contend that we have not tested every other experimental variable imaginable (e.g., different chemoattractants, different gradients, different cells, different basement membranes, etc.) would seem to hold us to a standard of negating every – and any flawed experiment or interpretation published in the literature. We have no problem with softening the text to read "highly dependent" rather than 'completely dependent', but we do not wish to detract from the fact that the notion that cancer cells can simply switch from protease-dependent to an equally effective protease-independent mode of invasion is not supported by the data, at least with regard to a physiologic barrier.

2. It is not clear to me that the authors have supported the claim that these BMs are “dominated by covalent cross-links” (line 114). This assay doesn’t speak to the presence of weak bonds, if any, so its not clear whether one modality is more dominant than the other. Further, if the authors wish to claim that “levels similar to those found in other highly cross-linked basement membranes”, they should include the quantification of the Western blot of percent NC1 in monomer vs. dimer form (as was done in the referred to manuscript – ref. 32).

As outlined in our last response, the literature is replete with works highlighting the importance of sulfilimine cross-links in maintaining the structural integrity of mechanically-loaded basement membranes in vivo. Nevertheless, the most commonly used basement

membrane surrogate, Matrigel, is devoid of covalent cross-links, let alone those derived from sulfilimines. However, as we pointed out previously, basement membranes do exist in vivo that are characterized by extremely low levels of sulfilimine cross-links, such as that found in the optic lens. Our only intent was to show that the basement membranes we use are replete with cross-links. As such, we have included the quantitation (78% dimer/22% monomer), but I would stress that the onus for performing this experiment should be placed on those investigators that claim evidence of protease-independent modes of basement membrane invasion as the absence of these cross-links would alter interpretation.

3. While the additional quantification of pore dynamics has been included, which is appreciated, it is not clear to me that the authors can conclude that the "pore dilation is primarily elastic in nature". To show this what they would need is the pore-size before cell arrives, the pore size as the cell invades through, and the pore size after cell passes through. It looks like they just have two time points (during and after, but not before; Fig. 7D, and 7F), so while they can show some elastic recovery, it looks like there might also be some plastic deformation (Fig 7G). Unless the authors can get the full trajectory of the pore size dynamics, the claim should be moderated.

We apologize if we did not make this point more clear. As opposed to the two time point shown in Figs 7D and F, we then quantified changes in pore size in over 500 basement membrane perforations from time 0 through day 6 of macrophage transmigration in 3 independent experiments. Under these conditions, pore size, having expanded to a peak of approximately 9 μm^2 from 4 μm^2 then decreased back to 5 μm^2 – an 80% recovery of control basement membrane pore size. As we cannot rule out the possibility that a further decrease in pore size might have taken place with longer culture period, that the distribution of pore size might have been skewed by stable cell protrusions (e.g., Fig 8G) or even continued cell migration, we prefer to conclude that most of the observed change in pore size can be attributed to elastic recovery.

Bahr et al. have adequately addressed the critique of Reviewer #3. There are however additional issues that should be addressed:

1) the color images in this manuscript are striking, but it would seem that pseudo colored images should be used to insure that the images can be evaluated by those who are color-blind. The authors state in their rebuttal in regard to a question about overexposure of laminin and collagen IV images that "With regard to image intensity, we have purposefully adjusted the images carefully to avoid overexposure." Presenting such single color images in grey scale might help as to this reviewer's eye the images also appear to be overexposed.

In discussing this issue with the Editor, we have included a series of supplementary figures where the red and green channels have been separated. Further, with regard to overexposure, at least two issues are relevant here. First, please be advised that when viewing images generated through the entire explant, we are actually imaging through 2 separate basement membranes. Second, to address this concern raised earlier, we specifically included normalized fluorescence line intensity analyses to characterize changes in normal BM staining intensity as well as the characteristics of proteolyzed and non-proteolyzed pores (see Figures 2 and 7 as well as Supplementary Fig 3).

In addition, including in the legends explanation of what the various colors indicate would help as a reader is faced with beautiful images that are not readily interpreted.

Agreed.

2)Panels E and F of Figure 3 should be modified. As presented in the Figure and described in the legend, these panels are said to be data on expression of proteases, yet both panel E and panel F include one serine protease receptor. In panel F, aspartic and serine proteases are designated as cysteine proteases. Similarly in the text in line 183, the authors use "cathepsin" to designate a proteinase family involved in ECM remodeling. This is not accurate as the name cathepsin is used for three proteinase families: aspartic, cysteine and serine.

Corrected- thank you for noting this oversight.